# A dirigent family protein confers variation of Casparian strip thickness and salt tolerance in maize

Yanyan Wang[1,5], Yibo Cao[1,2,5], Xiaoyan Liang[1], Junhong Zhuang[1,3], Xiangfeng Wang [4], Feng Qin [1,3] & Caifu Jiang [1,3✉]

Plant salt-stress response involves complex physiological processes. Previous studies have shown that some factors promote salt tolerance only under high transpiring condition, thus mediating transpiration-dependent salt tolerance (TDST). However, the mechanism underlying crop TDST remains largely unknown. Here, we report that *ZmSTL1* (*Salt-Tolerant Locus 1*) confers natural variation of TDST in maize. *ZmSTL1* encodes a dirigent protein (termed ZmESBL) localized to the Casparian strip (CS) domain. Mutants lacking ZmESBL display impaired lignin deposition at endodermal CS domain which leads to a defective CS barrier. Under salt condition, mutation of ZmESBL increases the apoplastic transport of $Na^+$ across the endodermis, and then increases the root-to-shoot delivery of $Na^+$ via transpiration flow, thereby leading to a transpiration-dependent salt hypersensitivity. Moreover, we show that the ortholog of ZmESBL also mediates CS development and TDST in Arabidopsis. Our study suggests that modification of CS barrier may provide an approach for developing salt-tolerant crops.

[1] State Key Laboratory of Plant Physiology and Biochemistry, College of Biological Sciences, China Agriculture University, 100193 Beijing, China. [2] Key Laboratory of Forest Silviculture and Conservation of the Ministry of Education, College of Forestry, Beijing Forestry University, 100083 Beijing, China. [3] Center for Crop Functional Genomics and Molecular Breeding, China Agricultural University, 100193 Beijing, China. [4] National Maize Improvement Center of China, Department of Plant Genetics and Breeding, China Agricultural University, 100193 Beijing, China. [5] These authors contributed equally: Yanyan Wang, Yibo Cao. ✉email: cfjiang@cau.edu.cn

Salt stress is one of the widely spread abiotic stresses, which affects about 20 million hectares of cultivated lands all over the world, and the area of the salt-affected farmland is increasing year by year due to the drastic climate change as well as the irrigation-dependent cultivating practice[1,2]. Given the fact that most crops are glycophytes, the salinization of farmland has emerging as a major environmental stress restricting global agriculture[3–5]. Under salt-affected farmland, crops take up excessive cations and anions (mainly $Na^+$ and $Cl^-$), thereby leading to ion toxicity and impairments of various physiological processes. Meanwhile, high concentration of soil salts decreases osmotic potential of soil solution, and then causes osmotic stress[2]. The salt-stress-induced osmotic stress, ion toxicity, and various subsequent effects (e.g. oxidative stress) together decline the quantity and quality of agricultural products[6,7]. In order to secure the sustainability of global agriculture, there is an urgent need of investigating the mechanisms underlying crop salt tolerance.

During the long course of evolution, plants have evolved a variety of mechanisms to adapt the salted land on which they lives, including osmotic adjustment, regulation of ion content, activation of ROS scavenging mechanism, growth reprograming, and so forth[2]. In the last decades, a number of factors associating with a specific or multiple salt-tolerance processes have been identified[2,8–10]. The regulation of $Na^+$ transport at cellular, tissue and whole-plant levels are essential for plant salt tolerance. Various $Na^+$ transporters have been identified from different transporter protein families, for example, the $Na^+/H^+$ antiporter SOS1 pumps $Na^+$ out of the cell[7,11], the HKT1 family transporters retrieve $Na^+$ from the xylem vessels then promote shoot $Na^+$ exclusion[12–14], the HAK family $Na^+$ transporters are associated with the regulation of root $Na^+$ concentration and root-to-shoot $Na^+$ translocation[15–17]. These transporters together with their regulators form a complex network regulating the uptake, transport and compartment of $Na^+$[2,6,18,19]. Also, previous studies have shown that shoot $Na^+$ exclusion is substantially associated with the transpiring status. For instance, AtRBOHF-mediated ROS production promotes shoot $Na^+$ exclusion and salt tolerance only under high transpiring condition, thereby mediating transpiration-dependent salt tolerance (TDST)[20,21]. Up to this day, the exact mechanism of TDST remains largely unclear, especially in major crops (e.g. maize, wheat, and rice). As most crops are farmed under high transpiring environment, it is valuable to investigate the mechanisms underlying TDST in crops, thus to facilitate the breeding of crops with improved salt tolerance in agricultural relevant transpiring environments.

Casparian strips (CS) are lignin-based cell-wall modification in the root endodermis of vascular plants, which is a tight barrier blocking the non-selective apoplastic transport of solutes and water[22,23]. It has been shown that the formation of CS is a highly ordered process. In the early stage, EXO70A1-mediated exocytosis localizes the Casparian strip membrane domain proteins (CASPs) into the CS domain to form discontinuous protein scaffolds[24,25], which are then linked together to form a continuous CASPs scaffold with the action of CIF1/2-SGN3-SGN1 complex (a peptide-receptor like kinase-coreceptor complex)[26–28]. Meanwhile, other factors essential for the formation of CS are recruited to the CS domain (e.g. RBOHF, PER64, ESB1)[29], and then the monolignol oxidation and polymerization occurred to form Casparian strip[29,30]. During the maturation of endodermis, MYB36 directly regulates the expression of the main genes involving the formation of Casparin strip[31], and the dirigent family protein ESB1 is suggested to mediate bimolecular coupling during the lignin biosynthesis[32]. It has also been shown that the development of endodermal Casparian strip and suberin lamellae shows large plasticity in response to environmental clues[22,33,34]. For instance, the biosynthesis and degradation of endodermal suberin lamellae are modulated by various plant hormones (e.g., ABA and ethylene) in response to nutrient stresses[33], and the endodermal CS barrier integrity is constantly checked by two CIF peptides and its maturation can be accelerated by low-$K^+$ signaling[26,35,36]. It is noteworthy that previous studies have also shown that the endodermal CS barrier matures closer to the root apex under salt stress[33,37,38], and the natural diversities of salt tolerance in rice are likely associate with the variations of the CS barrier[34], indicating the association between the CS barrier and crop salt tolerance. However, as the factors regulating the development of Casprin strip in crops are largely unidentified, and the genetic basis underlying the natural variations of CS barrier in crops remains unknown, making it challenging to clarify the role of endodermal CS barrier in crop salt tolerance.

Maize is one of the major cereal crops, and soil salinization is one of the major abiotic stresses restricting maize production worldwide[39,40]. Therefore, there is an urgent need of investigating the mechanism underlying maize salt tolerance. Previous studies have shown that natural maize inbred lines displayed large diversities of salt tolerance[17,41], and the diversities were partially ascribed to the QTL regulating the shoot $Na^+$ exclusion[14,17,42]. In this work, we investigate maize salt tolerance by focusing on the identification and functional study of the genes underlying the natural variation of transpiration-dependent salt tolerance (TDST). We show that the inbred line CIMBL45 confers a transpiration-dependent salt hypersensitive phenotype, due to the loss of function of ZmSTL1. *ZmSTL1* encodes a dirigent family protein designated as ZmESBL, which likely regulates the formation of endodermal Casparian strip (CS) by mediating the lignin deposition at the endodermal CS domain. Under high salt (NaCl) condition, the mutants lacking ZmESBL increase the apoplastic transport of $Na^+$ across the endodermis to reach vasculature, and then lead to an increased root-to-shoot $Na^+$ transport via the xylem-based transpiration flow, thereby leading to a transpiration-dependent salt hypersensitivity. In addition, we show that ESBL (the ortholog of ZmESBL in Arabidopsis) also mediates CS formation and salt tolerance in transpiring plants, and reveal that the lignin-based Casparin strip but not the suberin lamella promotes the shoot $Na^+$ exclusion and TDST. Overall, we identify a gene underlying the natural variations of crop endodermal CS barrier, and demonstrate that the lignin-based CS barrier substantially promotes shoot $Na^+$ exclusion and salt tolerance under high transpiring environment, hence providing valuable knowledge for developing salt-tolerant crops.

## Results

**ZmSTL1 underlies the salt hypersensitivity of CIMBL45.** In this study, we aimed to investigate the genetic basis underlying the natural diversities of transpiration-dependent salt tolerance (TDST) in maize. We compared the salt sensitivity of ~200 maize inbred lines under high (50% relative humidity, RH) and low (95% RH) transpiring environments, subsequently identified an inbred line CIMBL45 showing a salt hypersensitive phenotype under 50% RH condition, but not under 95% RH condition (Fig. 1a–d). More specifically, the salt-grown CIMBL45 plants were yellowish and showed a significant decrease of leaf SPAD (soil plant analysis development) value under 50% RH condition, but not under 95% RH condition (Fig. 1a–d). By contrast, the salt-grown 3H-2 (a salt-tolerance inbred line) showed an undetectable change of leaf SPAD value under either high or low transpiring conditions (Fig. 1a–d). These results indicated that CIMBL45 showed a transpiration-dependent salt hypersensitive phenotype, thus provided a genetic material to investigate TDST in maize.

We next determined the genetic basis underlying the transpiration-dependent salt hypersensitivity of CIMBL45. We

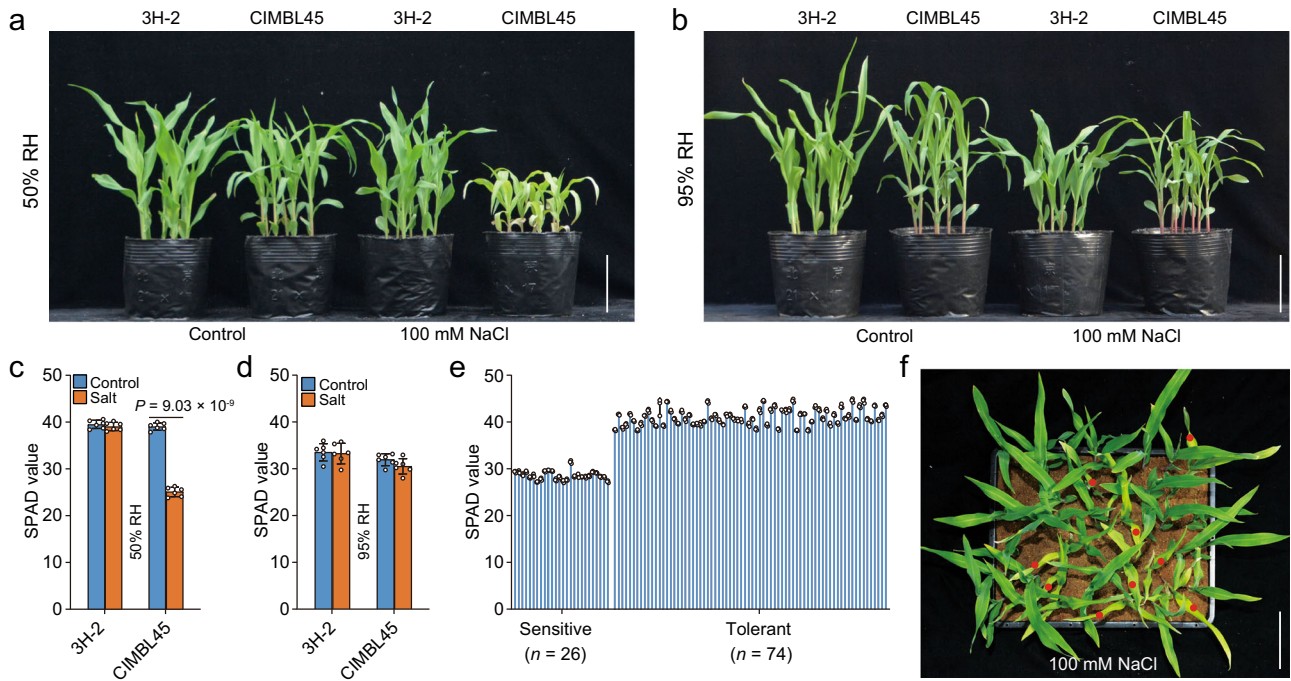

**Fig. 1 CIMBL45 shows a transpiration-dependent salt hypersensitive phenotype. a–d** The appearances (**a**, **b**) and SPAD values (**c**, **d**) of plants grown under control and salt (100 mM NaCl) conditions with indicated relative humidity (RH) (genotypes as indicated). **e**, **f** The SPAD values (**e**) and appearances (**f**) of the F2 plants grown under salt condition with 50% relative humidity. The results in (**c**) and (**d**) were means ± s.d. of three independent experiments. The SPAD values in **e** were expressed as means ± s.d. of three measurements. Statistical significance was determined by a two-sided *t*-test. Bars in (**a**), (**b**), and (**f**), 10 cm. Source data are provided as a Source Data file.

generated F1 seeds by crossing CIMBL45 and 3H-2, and then obtained the F2 population by selfing of the F1 plants. The F2 plants were grown under 100 mM NaCl condition with low (50%) atmospheric relative humidity to analyze the salt sensitivity. Subsequently, we observed that about 25% of the F2 plants showed a yellowish phenotype comparable with that of the CIMBL45 (Fig. 1e, f), indicating that the transpiration-dependent salt hypersensitivity of CIMBL45 is conferred by a single recessive gene. We designated the casual gene as *ZmSTL1* (*Salt-Tolerant Locus 1*).

**ZmSTL1 is a dirigent family gene**. We next identified the genetic identity of *ZmSTL1* by a combined application of bulked segregant analysis (BSA) based rough mapping and marker-based fine mapping. Firstly, we generated two DNA samples from the bulked salt-tolerant and salt-sensitive F2 progenies, and then produced 40-Gb whole-genome-sequencing data for each sample using Illumine Hiseq 3000 system (see methods). The subsequent ΔSNP index analysis mapped *ZmSTL1* to a 20-Mb region on Chromosome 1 (Fig. 2a) (see method). Secondly, we developed 11 markers (Supplementary Table 1), with which we mapped *ZmSTL1* to a 110-kb region using 876 salt-sensitive F2 progenies (Fig. 2b). There were two genes (*Zm00001d033941* and *Zm00001d033942*) within the 110-kb candidate region (Fig. 2b), with *Zm00001d033941* encoded a protein with unknown function and *Zm00001d033942* encoded a dirigent (DIR) family protein.

To validate the candidates, we used the CRISPR-Cas9-based approach to generate mutants lacking the candidate genes, and obtained two independent mutants for each candidate (Supplementary Figs. 1 and 2) (see Method). *Zm00001d033941*^crispr^*-1*, *Zm00001d033941*^crispr^*-2*, *Zm00001d033942*^crispr^*-1* and *Zm00001d033942*^crispr^*-2* respectively conferred a 7-, 2-, 10- and 23-bp InDel, all of which caused frameshifting (Supplementary Figs. 1, 2). We then analyzed the salt sensitivity of these mutants

(Fig. 2c–f), and unraveled that *Zm00001d033942*^crispr^*-1* and *Zm00001d033942*^crispr^*-2*, but not *Zm00001d033941*^crispr^*-1* and *Zm00001d033941*^crispr^*-2*, displayed a salt hypersensitive phenotype under 100 mM NaCl condition with 50% RH (Fig. 2c, e; Supplementary Fig. 3), suggesting that *Zm00001d033942* is the most likely candidate of *ZmSTL1*. Such a conclusion was supported by an allelism test, in which we took CIMBL45 and *Zm00001d033942*^crispr^*-1* alleles to be the loss-of-function alleles, and took the wild-type (WT) progenitor allele of *Zm00001d033942*^crispr^*-1* as the functional *Zm00001d033942* allele, then generated two F1 hybrids (CIMBL45/*Zm00001d033942*^crispr^*-1* and CIMBL45/WT). Follow-up analysis found that CIMBL45/*Zm00001d033942*^crispr^*-1* but not CIMBL45/WT was hypersensitive to salt stress under 50% RH condition (Supplementary Fig. 4). Moreover, we observed that *Zm00001d033942*^crispr^ mutants showed salt hypersensitive phenotype under 50% RH but not under 95% RH condition (Fig. 2c–f), which was similar to that of CIMBL45 (Fig. 1a–d). These results confirmed that the genetic variation at the dirigent family gene *Zm00001d033942* accounted for the transpiration-dependent salt hypersensitivity of CIMBL45.

**A 1-bp insertion truncates ZmSTL1 protein in CIMBL45**. We next identified the genetic variation underlying the functional variation of *ZmSTL1*/*Zm00001d033942*. We amplified and sequenced the genomic fragment covering *Zm00001d033942* (including 5'- and 3'- UTRs) using the primers 033942-G-F and 033942-G-R (Supplementary Table 1), subsequently identified 4 SNPs (SNP503, SNP507, SNP514 and SNP515) and one InDel (Ins512) variants between the DNA sequences of the 3H-2 and CIMBL45 (Supplementary Table 2), among which SNP503 leaded to an amino acid change from leucine to proline, and Ins512 (a 1-bp insertion) leaded to a frameshifting and truncation of *Zm00001d033942* in CIMBL45 (Supplementary Table 2; Fig. 2g).

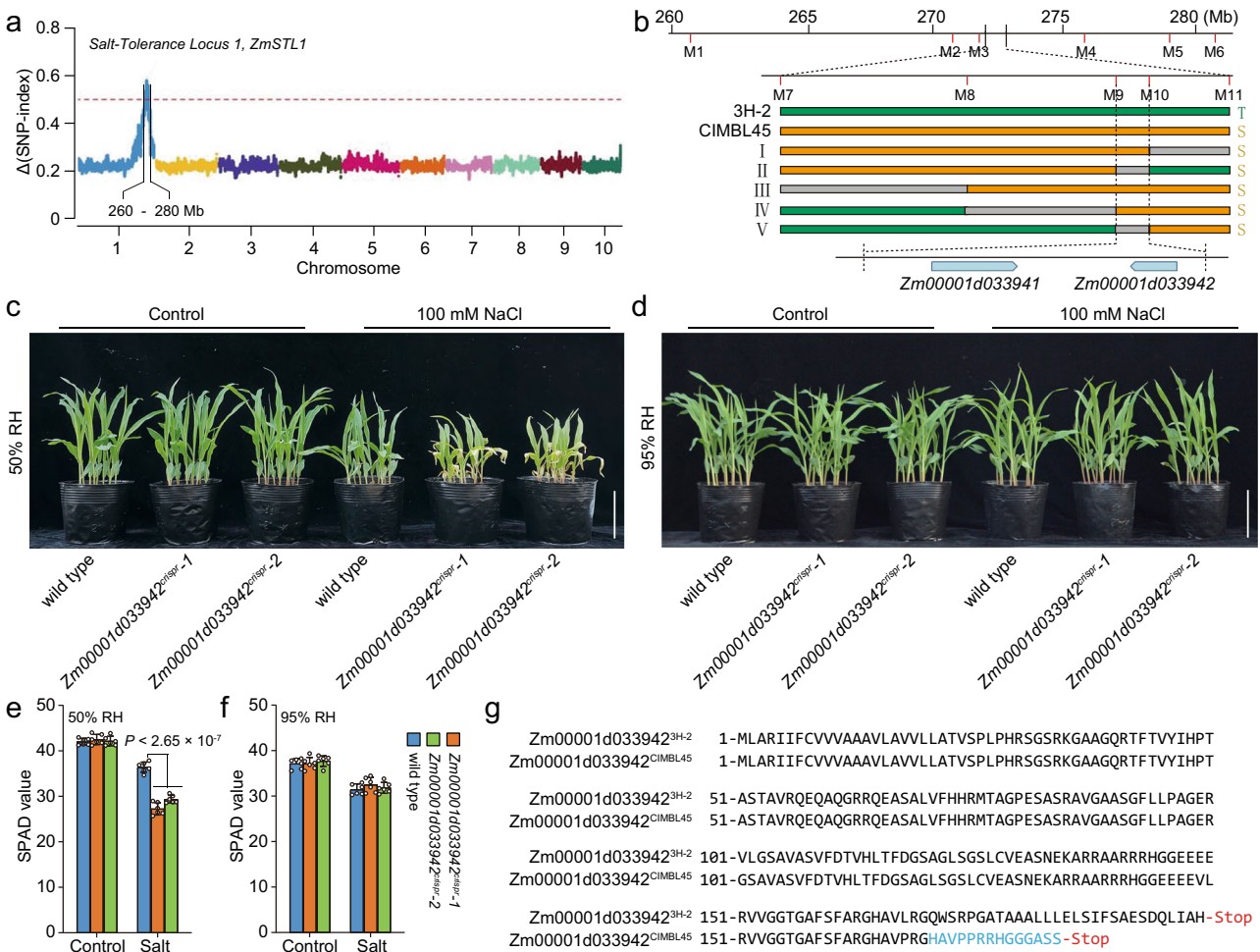

**Fig. 2 The dirigent family gene *Zm00001d033942* is the candidate of *ZmSTL1*. a** BSA-based analysis mapped *ZmSTL1* to a 20-Mb region on Chromosome 1 (see methods). (**b**) High-resolution linkage assay mapped *ZmSTL1* to a 110-kb region (see "Methods"). The graphical genotypes of five important recombinants were shown. Green, gray and yellow bars represented regions homozygous for 3H-2, regions with undetermined genotypes, and regions homozygous for CIMBL45, respectively. The two genes within the 110-kb candidate region were shown. **c–f** The appearances (**c, d**) and SPAD values (**e, f**) of *Zm00001d033942^crispr* and wild-type plants grown under control and salt (100 mM NaCl) conditions with indicated relative humidity (RH). **g** Alignment of the amino acid sequences of Zm00001d033942^3H-2 and Zm00001d033942^CIMBL45. The frameshifted sequence of Zm00001d033942^CIMBL45 was highlighted in blue. The results in (**e**) and (**f**) were means ± s.d. of three independent experiments. Statistical significance was determined by a two-sided *t*-test. Bars in (**c**) and (**d**), 10 cm. Source data are provided as a Source Data file.

As CIMBL45 conferred a functional null allele of *ZmSTL1/Zm00001d033942* (Fig. 2; Supplementary Fig. 4), we suggest that the frameshifting insertion (Ins512) is the most likely genetic variant underlying the functional variation of *ZmSTL1/Zm00001d033942* in CIMBL45, and the allele without Ins512 is the salt-tolerant allele of *ZmSTL1/Zm00001d033942*.

**ZmESBL is predominantly located in the Casparin strip domain**. *ZmSTL1/Zm00001d033942* belongs to the *DIR* gene family. The phylogenetic analysis of the DIR family proteins from maize (*n* = 33), rice (*n* = 20), soybean (*n* = 44), and Arabidopsis (*n* = 26) indicated that these proteins were grouped into five families (Family I–V), and Zm00001d033942 fell into Family V (Fig. 3a). Zm00001d033942, OS01G0879300, GLYMA_11G085200, GLYMA_05G242100, and AT5G42655 were grouped together to form a subfamily (Fig. 3a). As a previous study has designated AT5G42655 as Enhanced Suberin Like (ESBL) in Arabidopsis[31], accordingly, we named Zm00001d033942 as ZmESBL (*Zea may L.* Enhanced Suberin Like), and ZmSTL1/Zm00001d033942 is referred to as ZmESBL for the remainder of this manuscript.

The DIR family proteins have been shown to mediate regio- and stereoselectivity of bimolecular coupling during lignan and lignin biosynthesis[43], however, the physiological roles of ZmESBL and its orthologs remain unclear. Considering that previous studies have showed that *ESB1* (a Family V member) plays a role in the development of the endodermal Casparian strip (CS) barrier in Arabidopsis, and the loss-of-function of ESB1 leads to an ectopic deposition of suberin and lignin in endodermis[32], we then speculated that ZmESBL might also play a role in the regulation of endodermal CS development and/or suberin deposition. To address this hypothesis, firstly, we analyzed the tissue specificity of *ZmESBL* expression, and observed that *ZmESBL* mainly expressed in root tissue (Fig. 3b, c), and its transcripts were predominantly detected in the endodermis and the stele cells adjacent to the endodermis (Fig. 3d). Secondly, we determined the subcellular localization of ZmESBL by generating and analyzing the Arabidopsis plants expressing GFP-ZmESBL under the control of the *ESBL* (the ortholog of *ZmESBL* in Arabidopsis) promoter and the endodermal-specific *ESB1* promoter[32] (see methods). The results indicated that the GFP-ZmESBL was predominantly detected at the endodermal CS

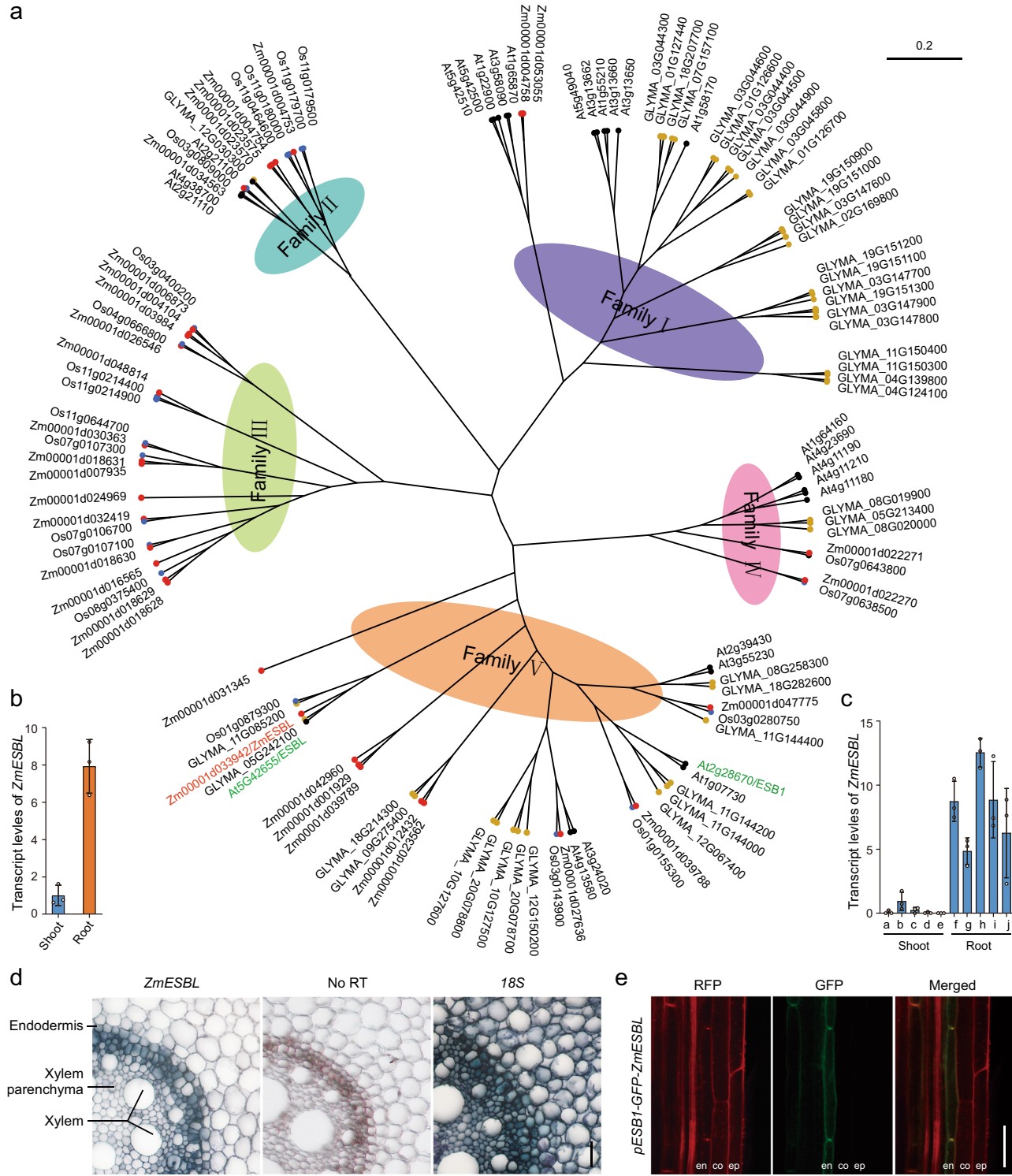

**Fig. 3 Phylogenetic analysis, expression pattern and the subcellular localization of Zm00001d033942/ZmESBL. a** The phylogenetic tree for maize, rice, soybean and Arabidopsis DIR proteins. **b** The transcript levels of *ZmESBL* in root and shoot tissues. **c** The transcript levels of *ZmESBL* in indicated samples. The data was downloaded from MaizeGDB a, 6–7 internode; b, 7–8 internode; c, leaf zone 1; d, leaf zone 3; e, mature leaf 8; f, root meristem zone; g, root elongation zone; h, root cortex; i, primary root; j, secondary root. **d** The in-situ RT-PCR assay. The dark blue signal indicated the presence of *ZmESBL* transcripts. No reverse transcription (No RT) and the *18S rRNA* (*18S*) provided controls. **e** The subcellular location of GFP-ZmESBL protein in the Arabidopsis plants transformed with *pAtESB1-GFP-ZmESBL*. The red fluorescence indicated the basic fuchsin staining of lignin. The results in (**b**) were means ± s.d. of three independent experiments. The images in (**d**) and (**e**) were representative of three independent repeats with similar results. Bars in (**d**) and (**e**), 20 μm. Source data are provided as a Source Data file.

domain of *pESBL-GFP-ZmESBL* and *pESB1-GFP-ZmESBL* plants (Fig. 3e; Supplementary Fig. 5). These observations suggest that ZmESBL likely plays a role in the regulation of endodermal Casparian strip formation and/or suberin deposition.

**Lacking of ZmESBL impairs lignin deposition at the CS domain**. We next compared the endodermal suberin deposition and Casparian strip formation between wild-type and *ZmESBL-crispr* plants. In order to make a proper comparison, firstly, we analyzed whether the loss-of-function of ZmESBL affects root development, and found that the primary root of the wild-type and *ZmESBL$^{crispr}$* plants showed comparable meristem size (~4.0 mm), elongation zone length (~3.0 mm), and a comparable length of the elongated endodermal cell (~125 μm) (Supplementary Fig. 6a–e), suggesting that ZmESBL unlikely regulates the aspects of root cell division and elongation. We then analyzed the pattern of the endodermal suberin deposition using a fluoral yellow 088 staining-based assay (see "Methods"). The results indicated that both the wild-type and *ZmESBL$^{crispr}$* plants started to form suberin lamellae in endodermis at ~5.0 cm from the root apex (~ the 344th elongated endodermal cell), and the intensities of the fluoral yellow 088 staining were comparable (Supplementary Fig. 7), indicating that ZmESBL is unlikely associated with the regulation of endodermal suberin deposition.

Previous studies have shown that the CS diffusion barrier is made of a lignin polymer[44]. We next compared the endodermal Casparian strip in the primary root of *ZmESBL$^{crispr}$* and wild-type plants using the basic fuchsin staining of lignin (see methods). We showed that, under control condition, the discontinuous lignin deposition at the endodermal CS domain started at 1.5 cm to the root apex (~ the 64th elongated endodermal cell) in the wild-type root, and then developed into ring-like continuous Casparin strips at 2.5 cm from the root apex (~ the 144th elongated endodermal cell) (Fig. 4a; Supplementary Fig. 6f). In contrast, the CS staining can be detected until at 2.5 cm from the root apex (~ the 144th elongated endodermal cell) in the *ZmESBL$^{crispr}$* root, and the fluorescence intensity was detected at significantly lower levels as compared with that of the wild-type control (Fig. 4a). We also showed that the control-grown and salt-grown root segments (2.0–2.5 and 2.5–3.0 cm from the root apex) of the *ZmESBL$^{crispr}$* mutants respectively displayed a ~15% and ~25% reduction of lignin content under control and salt conditions (Fig. 4b). Strikingly, we found that the Casparin strips of *ZmESBL$^{crispr}$* mutants were not fully continuous even at 3.0 cm from the root apex (~ the 184th elongated endodermal cell) (Fig. 4a). These observations indicated that lacking ZmESBL function leaded to defective Casparian strips with reduced and discontinuous lignin deposition. Moreover, we compared the CS structure of the wild-type and *ZmESBL$^{crispr}$* plants using a transmission electron microscopy assay, which permits us to analyze the CS structure in more detail. The results indicated that the wild-type root formed later Caparian strips that likely can function as an apoplastic barrier[33] at 2.5 cm from the root apex (~ the 144th elongated endodermal cell) (Fig. 4c), and the Caparian strip was located at the pericycle side of the endodermal cell-cell contact site (Supplementary Fig. 8a). In contrast, although *ZmESBL$^{crispr}$* can form Casparian strips at the similar position, the Casparian strips were incomplete and less organized (Fig. 4c; Supplementary Fig. 8b), which expanded to a wider domain at the elder root region (e.g. 4.0 cm from the root apex) (Supplementary Fig. 8c). Taken together, we conclude that ZmESBL plays an important regulatory role in maize CS development.

**Salt accelerates CS development by a ZmESBL-dependent manner**. Previous studies have observed that the development of the endodermal Casparian strips shows large plasticity in response to the environmental clues[22,33]. We next investigated the salt-induced plasticity of endodermal CS development and its association with ZmESBL. We observed that the discontinuous and continuous staining of CS lignin was respectively detected at 1.0 and 1.75 cm from the root apex (~ the 36th and 105th elongated endodermal cell) in the salt-grown wild-type root, being detected in younger cells as compared with that of the control-grown wild-type root (Fig. 4a). In addition, we observed by electron microscopy that the later Casparin strips of the salt-grown wild-type root were 25% wider than that of the control root (Fig. 4d). These results indicated that salt stress not only accelerates the development of Casparian strip, but also leads to the formation of a wider Casparian strip. By contrast, the salt-grown *ZmESBL$^{crispr}$* root conferred defective Casparian strips comparable with that of the control-grown *ZmESBL$^{crispr}$* root (Fig. 4a–c), suggesting that the salt-induced plasticity of Casparian strip is dependent upon the function of ZmESBL. Moreover, we observed that *ZmESBL* was mainly expressed in root tissue and its expression was induced 2–6 h following the onset of salt (100 mM NaCl) treatment (Fig. 4e). Notably, the root segments (up to 2.0 cm to root apex) where the onset and maturation of Casparian strip occurs displayed higher *ZmESBL* transcription (Fig. 4f), suggesting that the salt-induced reprograming of the endodermal CS development might be associated with the increased transcript levels of *ZmESBL*.

**Salt-induced strengthen of CS barrier is dependent on ZmESBL**. We next compared the function of the endodermal CS barrier in the primary roots of *ZmESBL$^{crispr}$* and wild-type plants using the apoplastic tracer propidiumiodide (PI). The results indicated that, in the root tip region (up to 2.0 cm from the root apex), PI penetrated the endodermis and entered into root stele both in *ZmESBL$^{crispr}$* and wild-type plants (Fig. 5a), indicated that the endodermal apoplastic barrier was not fully function. The PI penetration was completely blocked at about the 144th elongated endodermal cell (2.5 cm to root apex) of the wild-type root, by contrast, PI penetration was detectable even at 3.5 cm from the root apex (~ the 224th elongated endodermal cell) in *ZmESBL$^{crispr}$* root (Fig. 5a), indicating that loss-of-function of ZmESBL leads to a defective endodermal CS barrier. Moreover, we examined the effect of salt stress on the endodermal CS barrier (Fig. 5b). The results indicated that PI penetration was blocked at 2.0 cm from the root tip (~ the 127th elongated endodermal cell) in the salt-grown wild-type root (Fig. 4a–c), suggesting that the CS barrier become functional in younger endodermal cells under salt condition. By contrast, PI penetration was detectable at 3.5 cm from the root apex (~ the 264th elongated endodermal cell) in the salt-grown *ZmESBL$^{crispr}$* root, which was comparable with that of the control-grown *ZmESBL$^{crispr}$* root (Fig. 4a–c). Taken together with the observations that salt accelerates the development of Casparian strips and leads to the formation of wider Casparian strips, these results indicate that the salt-induced reprograming of CS development increases the function of endodermal CS barrier, and such a reprograming is dependent upon the function of ZmESBL.

**ZmESBL reduces the apoplastic loading of Na$^+$ into stele**. Shoot Na$^+$ exclusion is essential for crop adaptation to salt stress, and reducing the stele Na$^+$ loading provides a major mechanism of shoot Na$^+$ exclusion[2,6]. Considering that the endodermal CS barrier is able to block the excessive apoplastic transport of mineral elements across the endodermis to reach root stele[22], we then investigated the role of ZmESBL-mediated CS formation for preventing the stele Na$^+$ loading. We exposed the roots of the

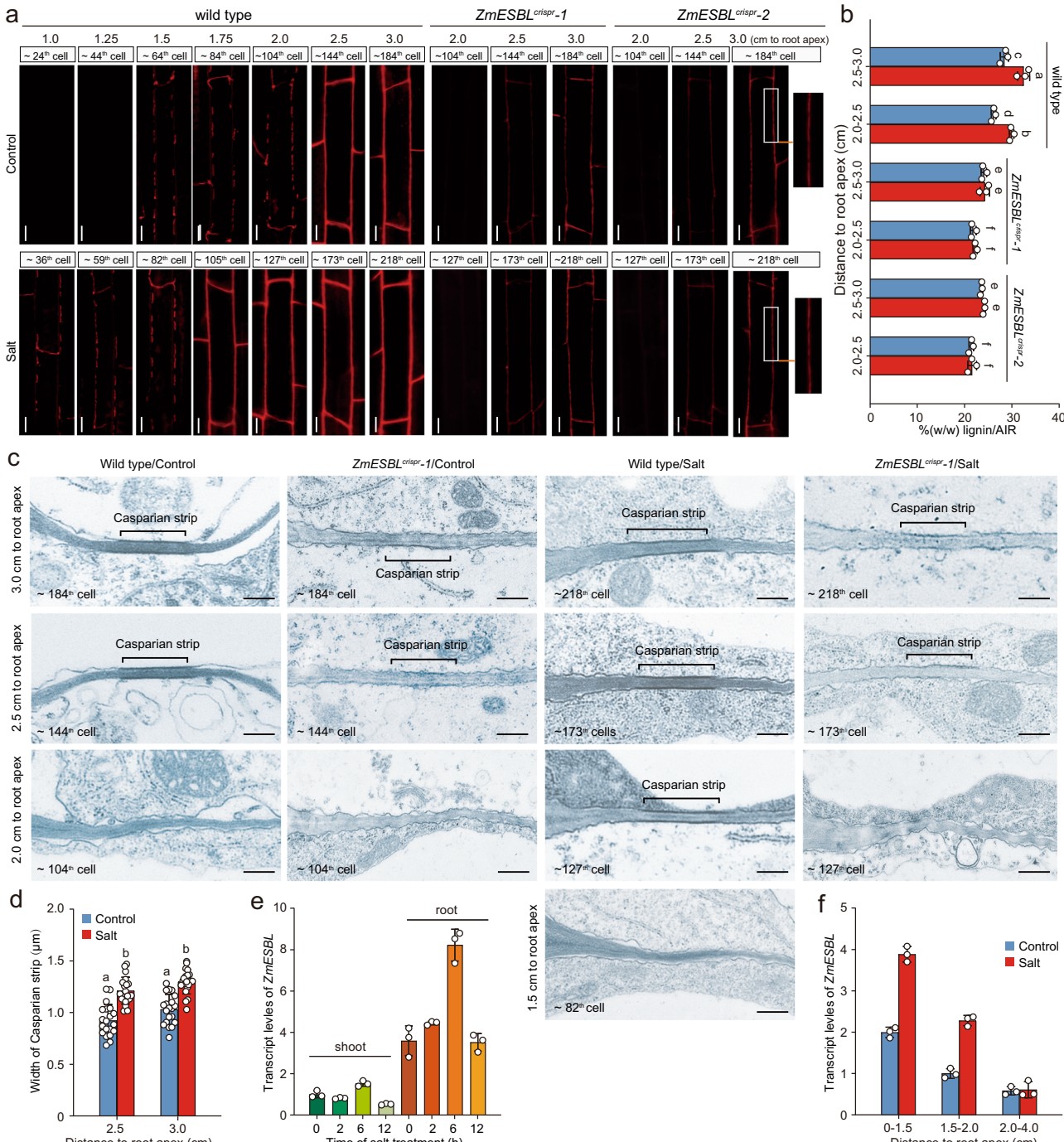

**Fig. 4 Lacking ZmESBL function leads to incomplete and less organized Casparian strips. a** Lignin staining-based observation of the endodermal Casparian strips in *ZmESBL^crispr* and wild-type plants grown under control and salt conditions. Basic fuchsin staining of lignin (see methods) were performed, showed the longitudinal view of the endodermal cells at 1.0 cm, 1.25 cm, 1.5 cm, 1.75 cm, 2.0 cm, 2.5 cm, and 3.0 cm from the root apex. **b** The lignin content of the wild-type and *ZmESBL^crispr* plants under the control and salt conditions. The plant were grown under control and salt (100 mM NaCl) conditions for 3 days, then the indicated root segments of the primary root were collected to measure the lignin content. **c** Transmission electron microscopy-based assay of Casparian strip structure in the wild-type and *ZmESBL^crispr*-1 plants under control and salt conditions. **d** The widths of the later Casparian strip in the wild-type roots under control and salt conditions. **e, f** The effect of salt stress on the transcript levels of *ZmESBL* in shoot and root tissue (**e**), or in indicated root segments with 6-h treatment (**f**). The results in (**b**), (**e**), and (**f**) were means ± s.d. of three independent experiments. Statistical significance was determined by a two-sided *t*-test. The images in **a** and **c** were representative of at least three independent repeats with similar results. Bars in (**a**) and (**c**), respectively, equaled to 20 and 0.5 μm. Source data are provided as a Source Data file.

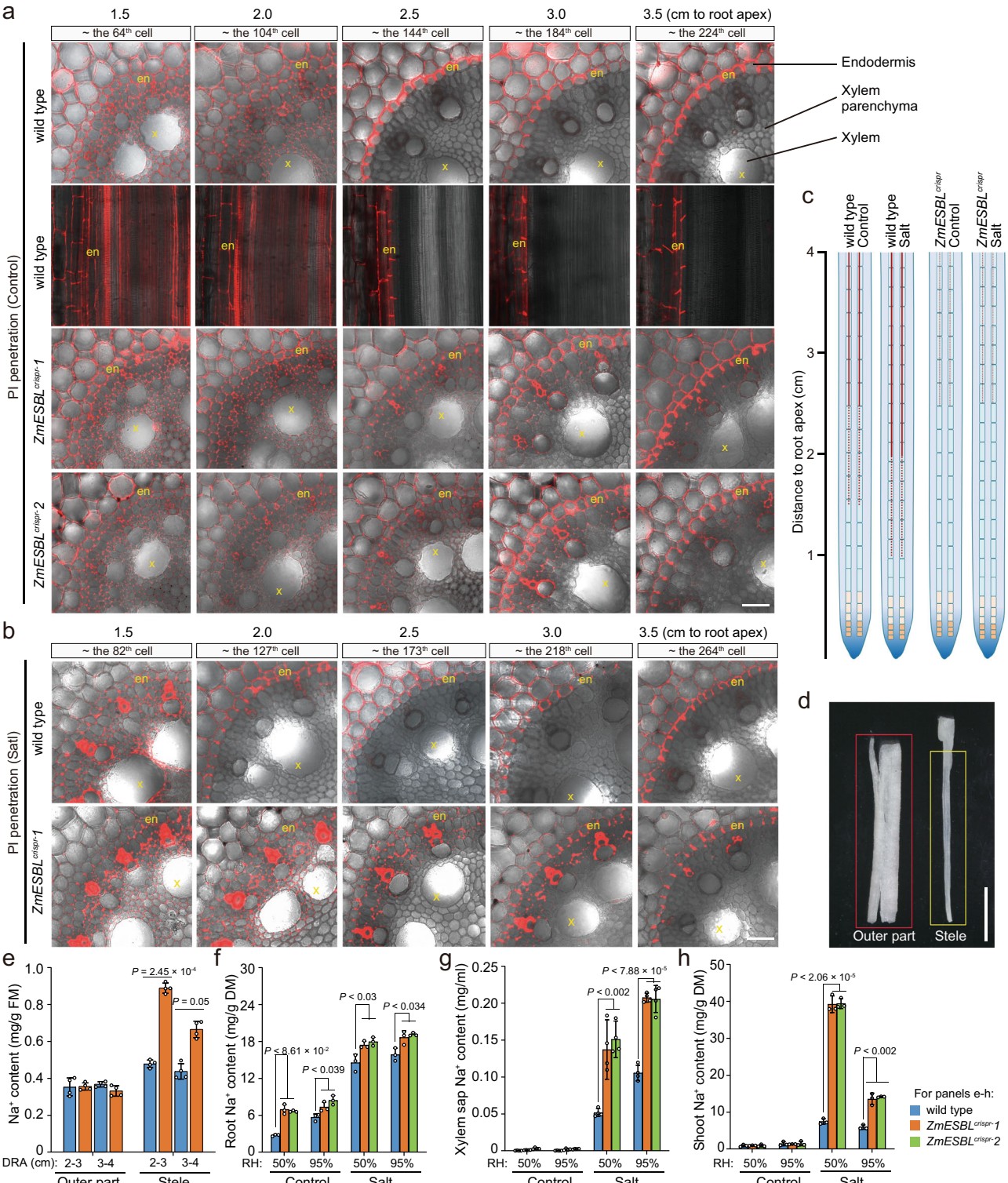

**Fig. 5 Lacking ZmESBL function leads to a defective endodermal CS barrier. a**, **b** PI penetration-based assay of the function of endodermal CS barrier in the primary roots of the plants grown under control (**a**) and salt (**b**) conditions (genotypes as indicated). PI penetration assay was performed at 1.5 cm, 2.0 cm, 2.5 cm, 3.0 cm, and 3.5 cm from the root apex (see methods). The wild-type roots under control condition were analyzed via both longitudinal and cross sections. en, endodermis; x, xylem. **c** A graphical demonstration of the function of the endodermal CS barrier between *ZmESBL*^*crispr*^ and wild-type plants. The red dash lines indicated Casparian strip, and the solid red lines indicated the fully functional CS barrier. **d** Demonstration of the separation of root stele from the outer part tissues. **e** The Na^+ contents in stele and outer part tissues of the indicated root segments. The roots of 3 days old seedling were treated with 200 mM Na^+ for 15 mins, and then the indicated root segments were collected and sampled as demonstrated in (**d**). The Na^+ contents were expressed as mg/g fresh mass (FM). DRA, distance to root apex. **f**–**h** Na^+ concentrations in the root (**f**), xylem sap (**g**) and shoot (**h**) of plants grown under indicated conditions. The images in (**a**) and (**b**) were representative of three independent repeats with similar results. The results in (**e**–**h**) were means ± s.d. of three independent experiments. Bars in (**a**) and (**b**), 40 μm. Statistical significance was determined by a two-sided *t*-test. Source data are provided as a Source Data file.

wild-type and $ZmESBL^{crispr}$ plants to salt solution (200 mM NaCl) for a short time (15 min), then collected the root segments 2.0–3.0 and 3.0–4.0 cm from the root apex (at where the CS barrier of $ZmESBL^{crispr}$ was less functional than that of the wild-type plants), and then separated the stele from the outer part tissues to analyze the $Na^+$ content (Fig. 5d). The results indicated that, while the outer part samples with different genotypes showed comparable $Na^+$ concentrations, the stele samples of $ZmESBL^{crispr}$ mutants showed higher $Na^+$ concentrations as compared with that of the wild-type plants (Fig. 5e), suggesting that the loss-of-function of ZmESBL increases the apoplastic transport of $Na^+$ across endodermis, thereby increasing the stele $Na^+$ loading.

**ZmESBL promotes shoot $Na^+$ exclusion in transpiring plants.** To further clarify the role of ZmESBL in regulating $Na^+$ delivery and salt tolerance, we compared the $Na^+$ concentrations in the root, xylem sap and shoot tissues of $ZmESBL^{crispr}$ and wild-type plants. The results indicated that the $Na^+$ contents in the root and xylem sap of the salt-grown $ZmESBL^{crispr}$ plants were respectively ~20% and ~50% higher than that of the wild-type plants, and the differences were identified both under 50% and 95% RH conditions (Fig. 5f, g). These observations supported the perspective that lacking ZmESBL function increases the appoplastic transport of $Na^+$ across endodermis to root stele and to be loaded into the xylem flow, and the effects were irrelevant to the transpiring status. Intriguingly, we found that the salt-grown $ZmESBL^{crispr}$ mutants accumulated 6 times more shoot $Na^+$ than the wild-type plants under high transpiring condition (50% RH), and the transpiration-restricting high RH (95%) reduced the accumulation by ~65% (Fig. 5h). These observations suggested that the disruption of ZmESBL function increases the apoplastic transport of $Na^+$ across endodermis, thereby leading to the excessive loading of $Na^+$ into root stele, nevertheless, the stele $Na^+$ can be excessively delivered to shoot and causes salt hypersensitivity only under high transpiring environment, which explains the transpiration-dependent salt hypersensitivity of $ZmESBL^{crispr}$ mutants. In addition, while previous studies have shown that $Na^+/K^+$ ratio is important for plant salt tolerance[6], and the impairment of Casparian strip is associated with a decreased tissue $K^+$ content[33,45], we found that the loss-of-function of ZmESBL leaded to a significant reduction of $K^+$ content under both control and salt conditions, and under both high and low humidity conditions (Supplementary Fig. 9a, b). Specifically, the salt-grown $ZmESBL^{crispr}$ plants under low RH (50%) condition conferred ~17% less shoot $K^+$ than that grown under high RH (95%) condition (Supplementary Fig. 9b), which probably also made a contribution to the transpiration-dependent salt hypersensitivity of $ZmESBL^{crispr}$ mutants.

**Arabidopsis ESBL regulates endodermal CS formation.** The orthologs of ZmESBL were identified in most of the higher plants, and ESBL is the ortholog of ZmESBL in Arabidopsis (Fig. 3a). We then determined whether ESBL also regulates the formation of the endodermal CS barrier. We generated two independent mutants lacking ESBL function (*esbl-1* and *esbl-2*) using the CRISPR-Cas9 based approach, with both of them conferred frameshifting mutations (Supplementary Fig. 10) (see method). Follow-up assay observed that the CS autofluorescence of the *esbl-1* and *esbl-2* mutants were significantly (~60%) weaker than that of the wild-type plants (Fig. 6a), and *esbl-1* and *esbl-2* showed a significant (~8%) reduction of root lignin content as compared with that of the wild-type plants (Supplementary Fig. 11). Meanwhile, the *esbl-1* and *esbl-2* mutants ectopically deposited a small amount of lignin at the cortex face of the endodermal cell-cell contact site. These observations suggest that lacking ESBL function impairs the lignin deposition at the normal CS domain (Fig. 6a). Nevertheless, the impairment was different from that of the *esb1* mutant, which deposited lignin in patches in the CS domain and displayed an ectopic deposition of lignin in the corners of the endodermal cells on both the pericycle and cortex faces[32] (Fig. 6a, b). In addition, while previous study have shown that *esb1* root displays ectopic suberin deposition in younger endodermal cells as compared with the wild-type root[32](Fig. 6c–e), we observed that the suberin deposition in the endodermis of the wild-type and *esbl* roots started at the same position (~ the 36th endodermal cells from the onset of elongation) (Fig. 6c), and the amount of suberin deposition in *esbl* root was comparable with that of the wild-type root (Fig. 6d, e). These observations suggest that ESBL and ESB1 regulate lignin deposition and CS formation via distinct mechanisms. We next compared the function of the endodermal CS barriers in *esbl-1*, *esbl-2*, *esb1* and the wild-type plants using the PI penetration assay (Fig. 6f, g). To quantify the barrier function, we counted the number of the endodermal cells from the onset of elongation to the site where the PI fluorescence was undetectable in root stele, and observed that the PI penetration was fully blocked at ~ the 13th endodermal cell from the onset of elongation in the wild-type root, but at ~ the 35th cell in *esbl-1*, *esbl-2* and *esb1* roots (Fig. 6g). These observations indicated that the function of the endodermal CS barrier was disrupted in *esbl-1* and *esbl-2* plants, which was comparable with that of *esb1* plants.

We next determined the mechanism by which ESBL orthologs regulate CS development. Firstly, as the above studies have shown that lacking of ZmESBL leaded to a reduction of lignin content (Fig. 4b), we then investigated the relationship between the Casparin strip defect and the reduction of lignin content in *esbl* plants, and found that the monolignol treatment (20 μM coniferyl alcohol plus 20 μM sinapyl alcohol) can neither restored the formation of Casparian strip nor rescued the function of the endodermal barrier of *esbl* mutant (Supplementary Fig. 12), suggesting that the Casparian strip defect of *esbl* is unlikely ascribed to the decrease of lignin content. Secondly, previous studies have shown that the exocytosis of CASP proteins into the CS domain to form discontinuous protein scaffolds occurs at the early (onset) stage of CS formation[24,25], and then form a continuous CASP scaffolds at the later stage[26–28]. We then investigated whether ESBL is associated with the formation of CASP scaffolds. To do this, we generated *pCASP1-CASP1-GFP* lines in the *esbl* and wild-type background, and observed that neither the formation of the early stage discontinuous CASP1 scaffolds nor the formation of the later stage continuous CASP1 scaffolds was affected in *esbl* mutant (Supplementary Fig. 13). These observations suggest that ESBL is unlikely associated with the onset of CS formation, hence suggesting that ESBL might be part of the machinery regulating the later CS formation processes.

**ESBL promotes salt tolerance in transpiring plants.** We further determined whether ESBL associates with the transpiration-dependent salt tolerance (TDST) by comparing the salt sensitivity and shoot $Na^+$ content of *esbl-1*, *esbl-2*, *esb1* and wild-type plants under the high and low transpiring conditions (Fig. 6h–j). We found that, while salt (150 mM NaCl) treatment leaded to an undetectable change of the leaf SPAD values of the wild-type plants under both 50% and 95% RH conditions (Fig. 6h, i), it caused a ~55% reduction of the SPAD values of *esbl-1* and *esbl-2* plants under 50% RH condition, but not under 95% RH condition (Fig. 6h, i), demonstrated that lacking ESBL function caused a transpiration-dependent salt hypersensitivity. *esb1* also displayed a transpiration-dependent reduction of leaf SPAD value following the onset of salt treatment,

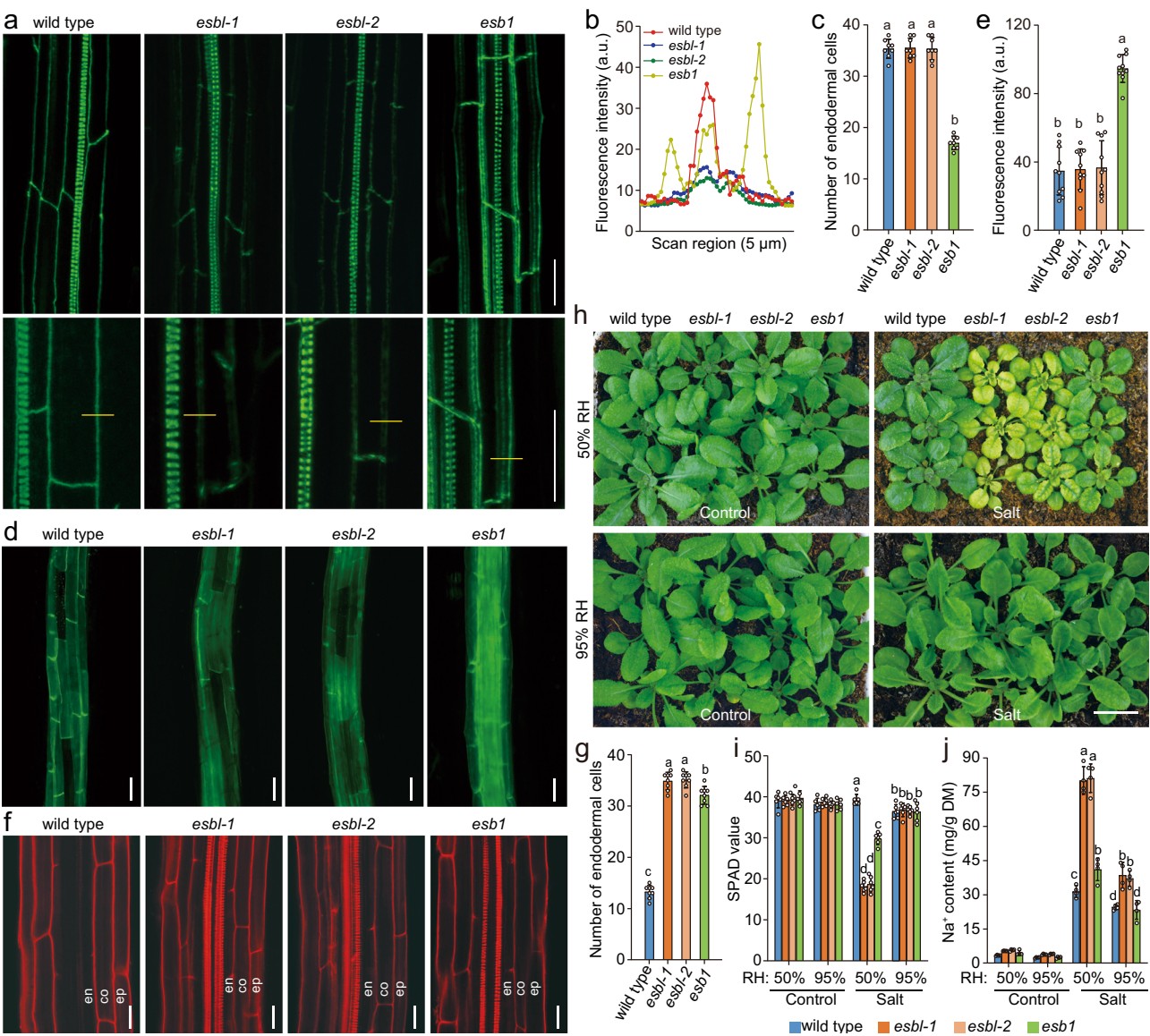

**Fig. 6 Arabidopsis ESBL is essential for CS development and salt tolerance under high transpiring condition. a** Confocal z-projections of endodermal Casparian strip autofluorescence. **b** The intensities and pattern of the CS autofluorescence in the indicated genotypes. The measurements were conducted at the locations indicated by the yellow lines in (**a**). a.u., arbitrary units. **c–e** Fluorescence assay of endodermal suberin deposition using fluorol yellow 088 staining. The results showed the locations where the endodermal suberin deposition start (**c**), the pattern (**d**) and intensities (**e**) of fluorescence in the indicated samples. **f–g** PI penetration-based assay of the function of endodermal CS barrier. ep, epidermis; co, cortex; en, endodermis. Quantification in (**c**) and (**g**) was done by counting endodermal cells after the onset of elongation as described in previous study[2], and the results were expressed as mean ± SD of 8 roots. The images in (**a**, **d** and **f**) were respectively taken at ~25th, ~40th and ~25th endodermal cell after the onset of elongation, and the images were representative of three independent repeats with similar results. **h–j** The appearances (**h**), SPAD values (**i**) and shoot $Na^+$ contents (**j**) of *esbl-1*, *esbl-2*, *esb1* and wild-type plants with the indicated treatments. Plants were grown in standard conditions for 4 weeks, and then watered once to soil saturation with either 150 mM NaCl or water (Control). Seven days later, the plants were photographed, the SPAD values of the 4th leaf were measured, and the shoot $Na^+$ contents were determined. The results in (**i**) and (**j**) were means ± s.d. of three independent experiments. Bars in (**a**, **d** and **f**), 20 μm; Bars in (**h**), 1 cm. Source data are provided as a Source Data file.

but the reduction was less apparent as compared with that of the *esbl-1* and *esbl-2* plants (Fig. 6h, i). We next observed that, under 50% RH condition, the salt-treated *esbl-1* and *esbl-2* plants showed a ~150% increase of shoot $Na^+$ content as compared with that of the wild-type plants, with *esb1* showed only a ~20% increase of shoot $Na^+$ content (Fig. 6j). We also observed that the transpiration-restricting high atmospheric relative humidity (95%) substantially reduced the shoot-$Na^+$ accumulation of the salt-treated *esbl-1* and *esbl-2* mutants (Fig. 6j). Taken together, we concluded that ESBL substantially confers shoot $Na^+$ exclusion and salt tolerance under high transpiring condition in Arabidopsis. Moreover, we observed

that *esbl* mutants showed 6–26% reduction of shoot $K^+$ content as compared with that of the wild-type plants, depending upon the growth conditions (Supplementary Fig. 14). Specifically, the salt-grown *esbl* mutants conferred ~26% less shoot $K^+$ than that of the wild-type plants under 50% RH condition (Supplementary Fig. 14), which likely also made a contribution to the transpiration-dependent salt hypersensitivity of *esbl* mutants.

**CS barrier confers salt tolerance in transpiring plants**. As above observations have shown that mutants lacking ESBL and ESB1

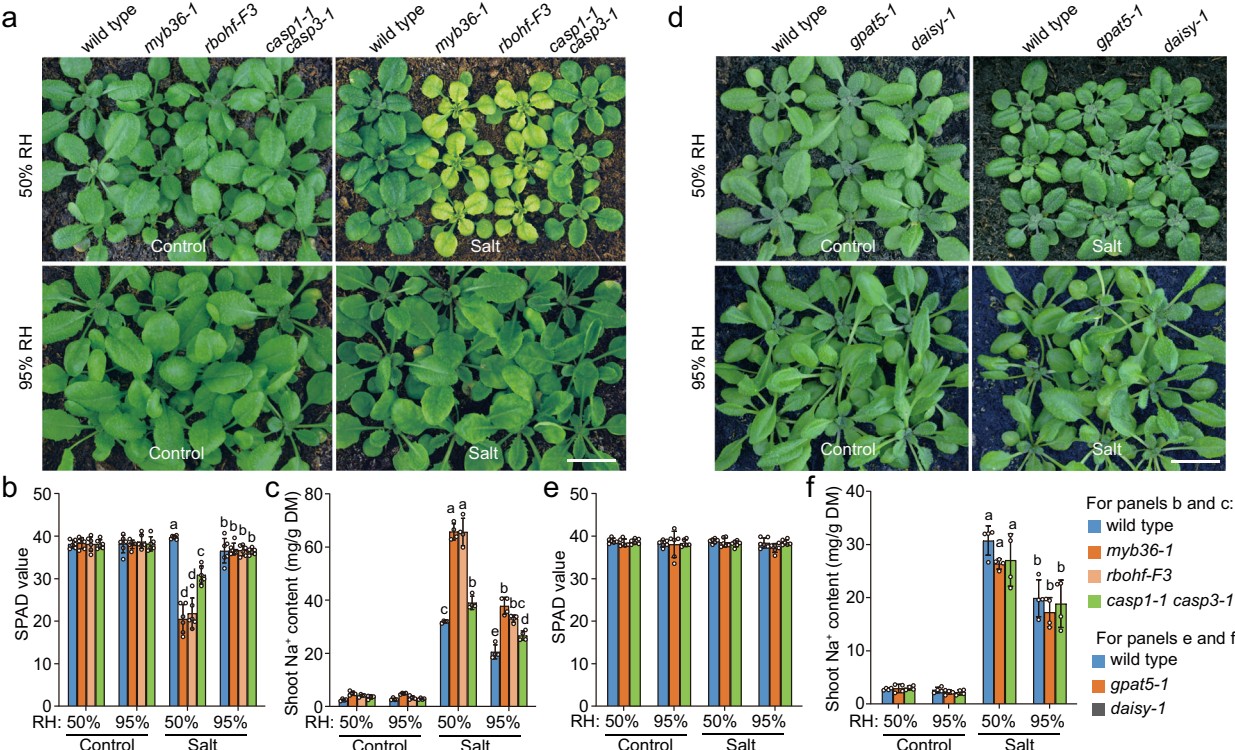

**Fig. 7 The lignin deposition at CS domain confers shoot Na⁺ exclusion and salt tolerance in transpiring plants.** **a–c** The appearances (**a**), leaf SPAD values (**b**) and shoot $Na^+$ contents (**c**) of *myb36-1*, *rbohf-F3*, *casp1-1 casp3-1* and wild-type plants with indicated treatments. **d–f** The appearances (**d**), leaf SPAD values (**e**) and shoot $Na^+$ contents (**f**) of *daisy-1*, *gpat5-1* and wild-type plants with the indicated treatments. Plants growth and salt treatment were as described in Fig. 6h–j. The results in (**b**, **c**, **e** and **f**) were means ± s.d. of three independent experiments. Bars in (**a**) and (**d**), 1.0 cm. Source data are provided as a Source Data file.

both conferred a defective CS barrier, it is probably a surprise to unravel that *esb1* is more tolerant to salt stress than *esbl* mutant under 50% RH condition (Fig. 6h–j). The discrepancies could be ascribed either to the ectopic suberization of endodermal cells, or to the compensatory lignin deposition at the endodermal cell-cell contact sites of *esb1* mutant[32]. To test these possibilities, we analyzed the salt sensitivities of additional CS formation-associated mutants, including *myb36-1*, *casp1-1 caps3-1* and *rbohf-F3*. Previous studies have shown that all of these mutants confer defective Casparian strips. In the meantime, these mutants show distinct pattern of ectopic suberin and/or lignin accumulation at endodermis, with *casp1-1 caps3-1* ectopically accumulates both suberin and lignin[25], *myb36-1* ectopically accumulates suberin but not lignin[31], *rbohf-F3* ectopically accumulates neither suberin nor lignin[30]. We found that *myb36-1* and *rbohf-F3* showed a comparable transpiration-dependent excessive shoot-$Na^+$ accumulation and salt hypersensitivity (Fig. 7a–c), by contrast, *casp1-1 caps3-1* was more tolerant to salt stress than *myb36-1* and *rbohf-F3* under 50% RH condition (Fig. 7a–c). These observations support the notion that the ectopic endodermal lignin deposition (in *casp1-1 caps3-1* and *esb1*) but not the ectopic endodermal suberin lamella deposition (in *myb36-1*) can partially complement the function of CS barrier in terms of promoting shoot $Na^+$ exclusion and salt tolerance. These discoveries also suggest that the transpiration-dependent salt hypersensitive phenotype of the plants lacking RBOHF function is substantially ascribed to the impairment of the CS barrier[21].

Furthermore, we analyzed the salt sensitivity of the mutants deficient of endodermal suberin lamella deposition (*daisy-1* and *gpat5-1*)[46,47], discovered that the salt-treated *daisy-1*, *gpat5-1*, and wild-type plants displayed comparable shoot $Na^+$ contents and salt sensitivity both under high and low transpiring conditions

(Fig. 7d–f), which supported the notion that the suberin lamella deposition was less relevant to shoot $Na^+$ accumulation and salt tolerance. Taken together with the observations that the mutants (*ZmESBL^crispr*, *esbl-1*, *esbl-2*, *myb36-1* and *rbohf-F3*) with a severe defect of lignin deposition in endodermal CS domain accumulated several times more shoot $Na^+$ and were hypersensitive to salt stress under high transpiring condition (Figs. 2c, 5h, 6h, 6j, 7a, 7c), these results demonstrated that the lignin-based endodermal CS barrier but not the endodermal suberin lamella conferred an important mechanism of shoot $Na^+$ exclusion and salt tolerance in transpiring plants.

## Discussion

The soil salinity is one of the major abiotic stress hindering the productivity and quality of agricultural products worldwide[2], thus there is an urgent need of characterizing salt-tolerant mechanisms that can be used to developing salt tolerant crops. Nevertheless, decades of efforts have made little progress, due to the complexity of salt response and its interaction with the variable environmental factors[2,8,20]. For this reason, the studies of salt tolerance using the plants growing in the physiologically realistic conditions are recommended. Previous studies have showed that the AtRBOHF-mediated ROS production confers salt tolerance under high transpiring conditions, mediating transpiration-dependent salt tolerance (TDST)[20,21]. As most crops are farmed under high transpiring environments, it is therefore valuable to investigate the molecular mechanism of crops TDST. Here we have shown that the dirigent (DIR) family protein ZmESBL underlies the natural variations of TDST in maize (Figs. 1 and 2). A 1-bp insertion causes the frameshifting and truncation of *ZmESBL* in the maize inbred line CIMBL45, and then leads to transpiration-

dependent salt hypersensitivity (Figs. 1, 2). *ZmESBL* provides an important genetic target for improving maize salt tolerance in the agricultural relevant transpiring environments.

DIR proteins were originally identified to regulate regio- and stereoselectivity of phenoxy radical coupling reactions[43], and the follow-up studies have revealed that the plant DIR proteins involve in a wide range of developmental processes and plant responses to various environmental signals. For instance, AtDIR6 is associated with secondary cell-wall formation[48]; AtDIR12 is likely associated with the synthesis of seed-specific neolignans[49]; AtDIR10/ESB1 is required for the formation of the endodermal Casparian strip (CS)[32]; a DIR-like protein PDH1 regulates soybean pod dehiscence likely by mediating lignin deposition[50]. In this study, we have observed that the DIR protein ZmESBL is associated with the lignin deposition at the endodermal CS domain in maize (Figs. 3e and 4a–c), the loss-of-function of ZmESBL leads to a defective CS barrier (Fig. 5a). In addition, we have shown that the Arabidopsis mutants lacking ESBL (the ortholog of ZmESBL) also confer a defective endodermal CS barrier due to the deficiency of lignin deposition (Fig. 6a, b, f, and g). Moreover, while previous studies have shown that the DIR protein ESB1 also involves in the regulation of endodermal CS formation in Arabidopsis, our study has revealed that ESBL and ESB1 regulate the CS development likely via distinct mechanisms. These discoveries provide an important demonstration that ZmESBL and its orthologs likely mediate a conserved mechanism of lignin deposition at the endodermal CS domain across monocot and dicot species, and across C4 and C3 species. In this regard, we showed that the loss-of-function of ESBL didn't affect the formation of the dot-like CASP1 scaffolds during the onset of CS formation (Supplementary Fig. 13), indicating that ESBL is less relevant to the onset of CS formation. Given the fact that the DIR family proteins have been shown to mediate regio- and stereoselectivity of bimolecular coupling during lignan and lignin biosynthesis[43], it remains highly possible that ESBL ortholog mediate bimolecular coupling during the polymerization of monolignol in the endodermal CS domain.

Sodium ($Na^+$) is the most abundant soluble cation in many saline farmlands, and shoot $Na^+$ exclusion is important for crop salt tolerance[12,13,51–53]. Our previous studies have shown that natural maize population displays large diversity of shoot $Na^+$ exclusion, which is partially ascribed to the variations of ZmHKT1- and ZmHAK4-mediated retrieving of $Na^+$ from the xylem flow, and the variation of ZmNSA1-mediated regulation of SOS1 activity[14,17,42]. Here we have shown that, under salt (NaCl) condition, loss-of-function of ZmESBL increases the apoplastic transport of $Na^+$ across endodermis to reach stele, then increases $Na^+$ content in xylem-based transpiration flow (Fig. 5e), and then causes excessive root-to-shoot delivery of $Na^+$ under high transpiring condition (Fig. 5g), which in turn leads to a transpiration-dependent salt hypersensitive phenotype. Taken together with the regulatory role of ZmESBL in CS development, these observations indicate that endodermal CS barrier provides a major mechanism preventing excessive stele-$Na^+$ loading, and that ZmESBL-mediated CS formation underlies natural variation of shoot $Na^+$ concentration and salt tolerance in transpiring maize plants. Intriguingly, it was previously shown that the functional variation of CS barrier is likely associated with the diversities of salt tolerance in rice[54,55], thus suggesting that the functional variation of CS barrier might underlies natural variation of salt tolerance in a wide range of crops. Nevertheless, such a perspective needs to be addressed in future study.

The maturation of endodermis cells involves the lignin deposition at Casparin strip domain at the early stage and the formation of suberin lamella at the later stage[22,23], with the lignin-based CS barrier prevents apoplastic delivery of minerals[56],

and the suberin lamella establishes a barrier for uptake of minerals from the apoplast into the endodermis[33]. Previous studies have shown that the maturation processes of endodermis are genetically regulated and influenced by environmental signals[33]. Previous studies have also shown that salt stress promotes maturation of the lignin-based Casparin strip and increases suberin lamella deposition in the endodermis cells[33,37,55], however, the molecular basis and the physiological significance of these processes are less understood. Here we have shown that salt stress accelerates the development of the lignin-based Casparin strips, as well as leads to the formation of wider Casparian strips. Intriguingly, the salt-induced reprograming of CS development is dependent upon the function of ZmESBL (Fig. 4a–c), more specifically, might be associated with the increase of *ZmESBL* transcription (Fig. 4e, f). Given the fact that the salt-grown $ZmESBL^{crispr}$ mutants accumulated 6 times more shoot $Na^+$ than that of the wild-type plants under high transpiring condition, we suggest that the endodermal CS barrier and its plasticity providing a major mechanism of shoot $Na^+$ exclusion and salt tolerance in transpiring plants. By contrast, the endodermal suberin lamella is less relevant to $Na^+$ hemostasis and salt tolerance (Fig. 7d–f). These observations also suggest that the physiological role of CS-mediated regulation of salts and probably other nutrients uptake should be determined in transpiring plants.

In summary, our study has identified *ZmESBL*, a dirigent family gene underlying the natural variation of salt tolerance under high transpiring condition. ZmESBL and its orthologs mediate a conserved mechanism of lignin deposition at the endodermal Casparian strip domain, with which we have demonstrated that the lignin-based Casparian strip barrier provides a major mechanism regulating stele-$Na^+$ loading, thereby promoting shoot $Na^+$ exclusion and salt tolerance in transpiring plants, suggesting that the genetic engineering of endodermal lignin deposition might provide a strategy for improving the salt tolerance of a wide range of crops.

## Methods

**Plant growth and salt treatment**. The maize inbred lines (3H-2 and CIMBL45) were obtained from National Maize Improvement Center of China at China Agricultural University. Pots with a height of 12 cm and a diameter of 10 cm were used for the growth of maize plants. In essence, the pots were filled with uniformly mixed Pindstrup substrate (www.pindstrup.com) and watered to soil saturation with water or salt (100 mM NaCl) solution. Ten seeds were planted in each pot, grown in the controlled environments with indicated atmospheric relative humidity for analyzing the salt sensitivity, measuring the leaf SPAD value, and collecting samples for the measurement of ion contents. For the growth and salt treatment of Arabidopsis plants, Seven-day-old plants grown on MS agar plates were transferred to pots filled with Pindstrup substrate, grown for 3 weeks in controlled environments (16/8-h light/dark cycle, 22 °C, ~50% relative humidity), then watered once to soil saturation with 150 mM NaCl solution or water (control). Seven days later, the plants were photographed, the SPAD values of the 4th leaf were measured, and the shoot samples were collected for the measurement of ion contents.

**Rough mapping and fine mapping of ZmSTL1**. The bulked DNA samples were extracted using DNAsecure plant kit (TIANGEN DP320-03), and then sequenced at Novagene, using the high-throughput sequencing platform highseq3000. The reads with more than 10% of missing bases and more than 50% of bases with Q-score lower than 20 were filtered, and the clean reads were mapped to the Maize reference genome (Zm-B73-REFERENCE-GRAMENE-4.0) using Burrows-Wheeler-Alignment Tool software (version 0.7.12)[57]. The alignment files were converted to SAM/BAM files for SNP-calling using SAMtools (version 1.11)[58]. SNP-index was calculated using host python and shell scripts. The SNPs detected in both bulked DNA samples were used for calculating Δ(SNP-index). Sliding window analysis was applied with 2-Mb window size and 10-Kb steps using BEDTools (version 2.29.2)[59], and Fig. 2A were generated using the R package CMPlot (version 3.7.0). Following the BSA-based rough mapping of *ZmSTL1*, we developed 11 markers, and used 876 salt hypersensitive F2 plants to conduct the fine mapping of *ZmSTL1*, and mapped *ZmSTL1* locus to a ~110-kb region between markers M9 and M10.

**Generation of the CRISPR/Cas9 knockout mutants.** The method of generating *ZmESBL*<sup>crispr</sup> mutants was generated using a CRISPR-Cas9 based approach[60]. A pCAMBIA-derived CRISPR/Cas9 binary vector with two gRNA expression cassettes targeting two adjacent sites of ZmESBL was generated and transformed into Agrobacterium strain EHA105, and then into the immature embryos of the maize inbred line 32990700 (wild type). In order to identify the positive mutants, the PCR products encompassing the gRNA-targeted sites for each of the transgenic plants were sequenced by capillary sequencing. To generate the CRISPR-Cas9 knockout mutants of ESBL in Arabidopsis, a pCAMBIA-derived CRISPR/Cas9 binary vector pHSE401–2gR with two gRNA expression cassettes was generated and transformed into Agrobacterium strain GV3101, and then transformed into Arabidopsis Col-0. The mutants were identified by amplifying and sequencing the genomic region encompassing the gRNA-targeted sites.

**Generation of *pESB1-GFP-ZmESBL* transgenic plant.** Approximately 1.5-kb DNA upstream of Arabidopsis *ESB1* was amplified from Col-0 genomic DNA using the primers pESBl-F and pESBl-R (Supplementary Table 1). The *GFP* coding sequence was amplified using the primers GFP-F and GFP-R (Supplementary Table 1). The ZmESBL coding sequence was amplified using the primers ZmESBL-F and ZmESBL-R (Supplementary Table 1). Those fragments were cloned into pGoldenGate-MCY2[61] to generate pESBl-GFP-ZmESBL construct, which was then introduced into Agrobacterium strain GV3101 and transformed into the Col-0 laboratory strain to generate *pESB1-GFP-ZmESBL* transgenic plant.

**Microscope observation of Casparian strip.** We used a lignin staining-based assay to observe endodermal Casparian strip in maize. In essence, the plants were grown under control or salt conditions for 4 days, then the cortex cells were carefully removed from the root with dissecting needles under stereoscope, and then the root was stained with the basic fuchsin solution (0.005% w/v in water; Sigma 857343) overnight at room temperature. The samples were rinsed in autoclaved deionized water for three times (5 min each) for observation. Confocal microscope setting for basic fuchsin was excitation 561 nm, and emission was 591–649 nm, and representative images are shown. The observation of Casparian strip in Arabidopsis was performed as follows. Five days old seedlings were incubated in solution containing 0.24 mol/L HCl and 20% methanol for 15 min at 57 °C, then incubated in solution containing 7% NaOH and 60% ethanol for 15 min at room temperature, then rehydrated by sequential incubating in 40%, 20%, 10%, 5% ethanol solutions (5 min each), and then keep in 25% glycerol for microscope observation of Casparian strip. Confocal microscope setting was excitation 488 nm, and emission was 493–537 nm. To obtain the z-stack images of Casparian strip, 0.95-μm step-size confocal microscope images were taken, and then radial optical sections were constructed. For the transmission electron microscope assay of Casparian strip[30], the root samples were fixed for 1 h at room temperature in fixative containing 2.5% (v/v) glutaraldehyde and 2% (v/v) formaldehyde in 0.05 mol/L phosphate buffer (PB; pH6.8), and then post-fixed for 1 h in 2% (w/v) osmium tetroxide in PB. Then, the samples were washed two times with distilled water, dehydrated through a gradient series of ethanol, infiltrated with Spurr's embedding medium, and then polymerized for 48 h at 60 °C. Ultrathin sections (100 nm) were prepared from the embedded samples and observed with the transmission electron microscope (Hitachi HT7800).

**PI penetration assay and suberin staining.** PI penetration assay of the barrier function of maize Casparian strips was conducted as follows. The root of three days old maize seedlings were incubated in the staining solution containing (2 μg/mL PI) (Sigma P4170) for 24 h, then rinsed in water twice and transversely sliced the sample, the sections were analysis with a Zeiss LSM710 confocal microscopy (excitation 561 nm, emission 591–649 nm). The barrier function of Arabidopsis Casparian strips is quantified using the PI penetration assay[56]. The endodermal suberin deposition was determined using fluoral yellow 088 staining[62].

**Determination of Na$^+$ and K$^+$ content.** Na$^+$ contents in shoot, root and xylem sap were determined using a 4100-MP AES device (Agilent, Santa Clara, CA, USA). In order to measure the Na$^+$ and K$^+$ contents in the root and shoot tissues, samples were oven-dried at 80 °C to constant weight, then incinerated at 300 °C for 3 h and 575 °C for 6 h. The ashes were dissolved in 10 mL 1% hydrochloric acid, diluted with an appropriate volume of 1% hydrochloric acid, and then the ion contents were measured. For the analysis of ion content in the xylem sap, 2-weeks-old seedlings were de-topped with a sharp blade, the xylem sap exuding at the cut surface was collected by a micropipette every 15 min for 1 h, and then the ion contents were measured.

**Generation of the phylogeny tree.** The phylogeny tree of DIR proteins was generated using Clustalx 2.1[63]. The sequences of Arabidopsis, maize, rice and soybean DIR proteins (accession numbers as shown in Fig. 3a) were downloaded from TAIR (https://www.arabidopsis.org/), MaizeGDB, RAP (https://rapdb.dna.affrc.go.jp/) and SoyBase, respectively.

## Data availability

The genome sequencing data generated in this study have been deposited in the NCBI database under BioProject number PRJNA706836 and BioSample number SAMN18146009 and SAMN18146011. All the raw images have been deposited to the BioImage Archive in BioStudies database under accession code S-BIAD390. Source data are provided with the paper. Source data are provided with this paper.

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

## Acknowledgements

The authors thank Xiangdong Fu and Wenkun Zhou for stimulating discussions. The authors acknowledge financial support from Beijing Outstanding Young Scientist Program (BJJWZYJH01201910019026 awarded to F.Q.), and the National Natural Science Foundation of China (Grant 32071933 and 31470350 awarded to C.J.).

## Author contributions

Y.W., Y.C., X.L., F.Q., X.W., and C.J. designed the experiments. Y.W. conducted the fine mapping of ZmSTL1, and worked together with Y.C. and J.Z. to generate the CRISPR/Cas9 knockout lines, and to study the function of ZmSTL1/ZmESBL and ESBL. X.L. and X.W. carried out the bioinformatics analysis. Y.W. and C.J. wrote the manuscript, with the other authors' contributions.

## Competing interests

The authors declare no competing interests.
