## [Peer Review File · Nature Communications]

A DIR family protein confers natural variation of Casparian strip and salt tolerance in maizeReviewers' Comments:

Reviewer #1:

Remarks to the Author:

The manuscript by Wang et al. presents a masterful characterization of the genetic basis for salt tolerance in maize and Arabidopsis. Through extensive and well carried out genetic studies, the authors identify a natural genetic variant in a DIRIGENT encoding gene that confers sensitivity to salt stress in maize under the physiologically relevant transpiring condition. The authors confirm the identity of the genetic variant and provide an anatomical explanation for the defects in salt exclusion. The gene identified, ZmESBL, exhibits specific localization to the Casparian strip (when expressed in Arabidopsis) and the mutants of this gene have defects in blocking the apoplastic movement of sodium into the root vasculature. Under transpiring conditions, sodium accumulates in the shoot leading to the salt-tolerance defects. The authors go even further and show that the function of the gene is conserved in Arabidopsis and that defects in the Casparian strip rather than the suberin lamellae, confers tolerance to transpiration-mediated sodium accumulation.

In short, the paper is exceptionally well done and complete. My only concerns relate to grammatically errors that are sprinkled throughout the text and may require a language editor to address.

Minor comment:

The images in Figure 6a are not of sufficient resolution to establish the subcellular deposition pattern of lignin. Comparisons between mutants should be made with higher resolution images cropped to show individual cells.

Reviewer #2:

Remarks to the Author:

The work of Wang et al delves deeper into the role of the casparian strip in salinity tolerance than previous work, in both maize and Arabidopsis. It is long been stated in the literature that the casparian strip is vitally important in the exclusion of Na⁺ from the shoot and that poorly formed endodermal barriers (for instance in rice) are a key reason why some plants accumulate more Na in the shoot and are susceptible to Na toxicity symptoms under certain conditions.

Here, the authors discover using one of the groups well known and highly effective genetic diversity screens in maize, a gene linked to Na accumulation in shoots. They argue that this gene is responsible for lignin deposition (ESBL) in maize and through homology also find the equivalent identified but uncharacterised gene in Arabidopsis, and that they both influences casparian strip development. They then reason that the knockouts mutants accumulate more leaf sodium in shoots. The phenotype they observe, high shoot Na and chlorosis, is evident in highly transpiring conditions and they therefore use the term transpiration-dependent salt tolerance.

It is an interesting and well presented study (but sometimes narrow in focus, without prosecuting other potential underpinning components of the phenotype) that adds to our understanding of the casparian strip and its role in salt tolerance.

I have a number of questions that I think it would be the authors to address to tighten the study (in terms of experiments) or add additional perspectives (that can enrich the discussion).

Major

Fig 1. Three measurements. So there are only 3 data points? Bar graphs very outdated, please change to scatter plots or show individual data points in graph type of choice.

This needs to be done for every graph in every figure (bar graphs hide data).

A key finding of the study is that Casparian strip formation starts higher up in mutant plants. So ESBL is needed to regulate barrier development in younger cells, but then somehow when cells get older

they still know (without ESBL) to make a proper barrier? But then salt treatment increases ESBL expression in the entire root? So it is needed higher up in the root as well, but we see no phenotype? Specifically, L239 statement ESBL is essential for formation of functional endodermal CS barrier. This sounds incorrect. L283. How do the mutant plants make the barrier then (which the authors show), it just starts higher up, but it is there eventually. How is ESBL essential then? L306. Is the onset of CS formation delayed as shown earlier, or is the entire CS formation disrupted, as stated in this sentence here? Formation vs onset. Please can the authors elaborate on their hypothesis and fully explain the logic here.

Do the shoots develop toxic symptoms even in high humidity if the salt treatment is prolonged? Can you really justify call this transpiration dependent?

L249. Lack of ESBL increases Na⁺ transport. Previous studies (Pfister et al 2014, eLife, Barberon et al 2016 Cell) suggest that defective CS leads to loss of K⁺ and that this is the origin of the problems the plants have. The phenotype you should (chlorosis) is a classic K⁺ deficiency phenotype. It would be ideal if you also included a measure of K in the tissues.

L260. It is curious that there is not more Na⁺ in WT roots, as they exclude it from the transpiration stream. This is typically seen in roots of excluders. Why not here?

L269. Even worse, now there is more Na⁺ in mutant roots? Where does all the Na⁺ go that WT excludes from shoot? It is shown that the more Na in mutant is in stele (Fig 5e), but why is there not more in WT in cortex if it is excluded by the endodermis? Where does it go? Washed out during preparation? Why did it not get washed out of xylem vessels in the mutant then in the stele preparation (also apoplast)?

L192. Speculation that ZmESBL is important for Casparian strip formation, as the first thought. But this makes no sense to me. Why not suberin? Why look at lignin as the first choice?

L210. The use of the lignin stain as showing less lignin quantity. This should be stated much more carefully, as it is not trivial to equal stain signal with lignin quantity. Can lignin actually be properly quantified in the tissue rather than just using image analysis through biochemical or other means. Is the quantification of lignin in the tissues robust enough?

L200, use of ESB1 promoter. Why not the ESBL promoter, the homologue of the ZmESBL? It makes little sense. In particular, as using the ESB1 promoter ensures expression of ZmESBL in endodermis only.

Paragraph starting L203. Clarification about cell length is needed here – it would be good to have a supp graph with the data for this paragraph.

L351, daisy mutants, I don't understand why the authors say the finding support their data/hypothesis. The daisy mutants are salt sensitive, but contain similar Na⁺ than wildtype, while they have defects in suberin deposition. How does this fit?

Fig 3e. Image not acceptable. Image stack needs to be taken, and maximum intensity projection shown that displays the "net" of the CS (like shown in many of the references they give). Is some form of co-staining required here too?

Minor

Reference numbering is not according to how they appear in the text, but numbers plus alphabetic, is this normal?

L196, no reference given for "previous study"

L246, what does it mean ESBL "enhances" maize endodermal CS function. Enhances? Just above it was essential for it?

L290. Extracting quantification data from staining again, need to be more carefully written

L297, images might suggest that esbl mutants have more passage cells? Is this the case or just by chance like this in the given images?

L317 esbl knockout has transpiration dependent phenotype. Arabidopsis esb1 was shown to be salt sensitive under standard growth condition. Do the authors suggest that salt phenotype of esb1 is also transpiration dependent?

L398. Na⁺ is the most abundant element in saline farmland. Is this a fact for each single saline farmland? Maybe re-write

Fig 1, please clarify what a SPAD value is, it is not explained anywhere in the manuscript.

Reviewer #3:

Remarks to the Author:

To understand transpiration-dependent salt tolerance mechanism in crop, the authors screened maize inbred lines and identified CIMBL45, which shows transpiration-dependent hypersensitivity of salinity. Through the mapping, sequencing analysis and knock out studies, the authors discovered that a DIR family gene, ZmESBL, was the causative mutation and further analyzed its molecular mechanisms involved in Casparian strip development. The authors provided significant advances in the mechanisms of salt tolerance in crops and suggested a novel mechanism of ESBL in the regulation of Casparian strip development. However, some critical experiments/points should be considered in order to solidify the conclusion in the manuscript.

1) Casparian strip structure in maize:

- It is necessary to clearly show the Casparian strip (CS) structure in maize, including how CS initiated in the early stages and matured in the later stage. CS of Arabidopsis exists in a ring-like structure, is it similar in maize? The CS of Arabidopsis starts with a dotted pattern and then develops in a continuous form through elongation. Is this similarity in maize? When and in what form does suberin deposition occur in maize? How do the structural properties of CS change after salt treatment? Fig4 images are not enough to understand the structure of CS.

- It is also necessary to more accurately describe the ko phenotype in both quantitative and qualitative terms.

- Interpreting CS development in response of salt treatment have to be careful. Since CS development is related to the differentiation of the roots, it is necessary to carefully observe whether the root development is affected by the salt or whether the root development is the same, but only the CS development is accelerated. CS position data should be analyzed after carefully observing changes in meristem zone size and elongation zone size.

- To verify CS permeability, PI was treated for 24 hours and then sectioned and observed. A control experiment for this PI method and quantitative validation is required. Lignin synthesis inhibitor, Lignin synthesis inhibitor+monolignol treatment can be used. To visualize permeability, longitudinal sections are required.

2) Molecular mechanisms of ZmESBL:

- The authors suggest that the action of ZmESBL is different from that of ESB1, but there is no suggestion as to what mechanism they think it is. Is it linked to a decrease in lignin production? Does the ectopic treatment of monolignol restore the ko phenotype? Or is it related to the CIF1/2-SGN3 pathway? Does ectopic treatment of CIF1/2 restore the ko phenotype? How about the localization of

CASP1 in ko? Further experimentation is needed to see how ZmESBL affects the already identified critical elements of CS development.

- The authors tried to understand the mechanism of ESBL using the Arabidopsis mutant study, which can be improved by more accurately describing the phenotype and adding quantitative assessments. "Weak" or "strong" is not enough to describe CS structure. It is necessary to describe the characteristics by observing the CS structure in a more magnified view (Fig 6a), and it is necessary to describe the suberin pattern in comparison with the development stage (Fig 6b). Both are missing quantitative analysis.

3) Minor points:

- Description of SPAD is missing.

- Localization of ZmESBL (Fig 3e): It is good to clearly show correlation of Casparian strip.

Additionally, using PM reporter protein with ZmESBL helps readers understand its localization pattern.

- In 75, 214 lines, the author said that CS shows large plasticity, but since the degree and range of plasticity of CS and suberin are different, it is necessary to distinguish and explain it.

- In this paper, the roles of CS lignin and suberin are distinguished, and the importance of CS is especially emphasized. The characteristics of the two and the molecular mechanism of their action need to be explained.

The following outlines our specific responses to the reviewers' comments:

Reviewer #1 (Remarks to the Author):

The manuscript by Wang et al. presents a masterful characterization of the genetic basis for salt tolerance in maize and Arabidopsis. Through extensive and well carried out genetic studies, the authors identify a natural genetic variant in a DIRIGENT encoding gene that confers sensitivity to salt stress in maize under the physiologically relevant transpiring condition. The authors confirm the identity of the genetic variant and provide an anatomical explanation for the defects in salt exclusion. The gene identified, ZmESBL, exhibits specific localization to the Casparian strip (when expressed in Arabidopsis) and the mutants of this gene have defects in blocking the apoplastic movement of sodium into the root vasculature. Under transpiring conditions, sodium accumulates in the shoot leading to the salt-tolerance defects. The authors go even further and show that the function of the gene is conserved in Arabidopsis and that defects in the Casparian strip rather than the suberin lamellae, confers tolerance to transpiration-mediated sodium accumulation.

Our response: We thank the reviewer for these positive comments on our submission.

In short, the paper is exceptionally well done and complete. My only concerns relate to grammatically errors that are sprinkled throughout the text and may require a language editor to address.

Our response: Thanks the reviewer for the comments. We have carefully looked through the manuscript to correct grammatical errors.

Minor comment:

(1)The images in Figure 6a are not of sufficient resolution to establish the subcellular deposition pattern of lignin. Comparisons between mutants should be made with higher resolution images cropped to show individual cells.

Our response: Thanks the reviewer for this valuable comment. In the revised manuscript, we made comparisons between mutants and wild-type plants with higher resolution images, photos that show Casparin strip at individual cells have been added (**Figure 6a**).

Fig. 6a Confocal z-projections of endodermal Casparian strip autofluorescence.

Reviewer #2 (Remarks to the Author):

The work of Wang et al delves deeper into the role of the casparian strip in salinity tolerance than previous work, in both maize and Arabidopsis. It is long been stated in the literature that the casparian strip is vitally important in the exclusion of Na⁺ from the shoot and that poorly formed endodermal barriers (for instance in rice) are a key reason why some plants accumulate more Na in the shoot and are susceptible to Na toxicity symptoms under certain conditions.

Here, the authors discover using one of the groups well known and highly effective genetic diversity screens in maize, a gene linked to Na accumulation in shoots. They argue that this gene is responsible for lignin deposition (ESBL) in maize and through homology also find the equivalent identified but uncharacterised gene in Arabidopsis, and that they both influences casparian strip development. They then reason that the knockouts mutants accumulate more leaf sodium in shoots. The phenotype they observe, high shoot Na and chlorosis, is evident in highly transpiring conditions and they therefore use the term transpiration-dependent salt tolerance.

It is an interesting and well presented study (but sometimes narrow in focus, without prosecuting other potential underpinning components of the phenotype) that adds to our understanding of the casparian strip and its role in salt tolerance.

I have a number of questions that I think it would be the authors to address to tighten the study (in terms of experiments) or add additional perspectives (that can enrich the discussion).

Major

(1) Fig 1. Three measurements. So there are only 3 data points? Bar graphs very outdated, please change to scatter plots or show individual data points in graph type of choice. This needs to be done for every graph in every figure (bar graphs hide data).

Our response: Thanks the reviewer for this valuable comment. We have changed all the bar graphs to scatter plots.

(2) A key finding of the study is that Casparian strip formation starts higher up in mutant plants. So ESBL is needed to regulate barrier development in younger cells, but then somehow when cells get older they still know (without ESBL) to make a proper barrier? But then salt treatment increases ESBL expression in the entire root? So it is needed higher up in the root as well, but we see no phenotype? Specifically, L239 statement ESBL is essential for formation of functional endodermal CS barrier. This sounds incorrect. L283. How do the mutant plants make the barrier then (which the authors show), it just starts higher up, but it is there eventually. How is ESBL essential then? L306. Is the onset of CS formation delayed as shown earlier, or is the entire CS formation disrupted, as stated in this sentence here? Formation vs onset. Please can the authors elaborate on their hypothesis and fully explain the logic here.

Our response: Thanks the reviewer for these very important comments. We have conducted multiple additional experiments to address these comments:

(a) We have determined the endodermis cell number from the first fully elongated cells to the site where starts to form early (discontinuous) and later (continuous) Casparian strip (**Supplementary Figure 6; Figure 4a-c**), the results were in consistent with our previous conclusion that the Casparian strip formation starts higher up in *ZmESBL^{crispr}* mutant plants.

(b) We have compared the Casparian strip of the wild-type and *ZmESBL^{crispr}* mutant plants using transmission electron microscopy and additional confocal microscopy assay, these studies also showed that the mutants confer incomplete and less organized Casparian strips even in late stage (3.0 cm to the root apex) endodermis cells, featured by less lignin deposition and discontinuous of the Casparian strips (**Figure 4a-c**).

(c) We have made a detail analysis of the effect of salt stress on CS onset, formation and structure, and observed that the Casparian strip initiated early (the analysis have taken the size of meristem zone, elongation zone, and the length of the cells in consideration) (**Figure 4a and c**), and that the salt stress also increase the width of the later CS barrier by 20% in wild-type plants (**Figure 4d**).

(d) We have showed the Casparian strip formation is dependent upon the function of *ZmESBL* under both control and salt conditions (**Figure 4a-c**), and we have compared the transcript levels of *ZmESBL* in different root segments (0-1.5; 1.5-2.0; 2.0-4.0 cm to root apex) under control and salinity conditions. The results indicated that the younger root segments (up

to 2.0 cm to root apex), where the onset and maturation of Casparian strip occurred, displayed higher *ZmESBL* transcription that can be further increased by salt stress (**Figure 4f**), suggested the role of *ZmESBL* in the reprogramming of Casparian strip under salt stress condition.

(e) To address the exact role (onset versus formation) of ESBL orthologs, we generated a pCASPI-CASPI-GFP in *esbl* mutant background, and observed that neither the formation of the discontinuous CASPI scaffolds at the onset stage nor the formation of the continuous CASPI scaffolds at the later stage of CS formation was affected in *esbl* mutant (**Supplementary Figure 13**). These observations suggested that ESBL unlikely involves in the onset of CS formation, hence suggested that ESBL might be part of the machinery that is recruited to the CS domain to mediate a later process of CS formation. Given the fact that the DIR family proteins have been shown to mediate regio- and stereoselectivity of bimolecular coupling during lignan and lignin biosynthesis, it remains highly possible that ESBL ortholog mediate bimolecular coupling during the polymerization of monolignols.

Supplementary Figure 13 The subcellular localization of CASPI-GFP in the wild-type and *esbl-1* background. The discontinuous (a) and continuous (b) CASPI-GFP were observed in both wild-type and *esbl-1* plants, and were observed at the same root developmental stage reflecting by the number of endodermal cells from the onset of cell elongation (c,d). Bars = 20 μ m.

Supplementary Figure 6 The profiles of maize primary root under salt and control conditions. **(a)** A graphical demonstration of the profiles of the maize primary root grown under control condition. The red dash lines indicated Casparian strip, and the solid red lines indicated the fully functional CS barrier. **(b)** A demonstration of the cell size at indicated developmental stages. Calcofluor White stains cellulose. **(c-e)** The size of meristem **(c)** and elongation **(d)** zone, and the length of the fully elongated endodermal cell **(e)**. The genotypes and treatments were as indicated, and the result indicated that the salt (100 mM NaCl) reduced the size of the meristem by $\sim 500 \mu\text{m}$, reduced the length of the elongation zone by $\sim 500 \mu\text{m}$, and reduced the length of the elongated endodermal cells by 12%. **(f)** Longitudinal and cross section view of the Casparin strips at the early (2.0 cm to root apex; \sim the 104th elongated endodermal cell) and later (3.0 cm to root apex; \sim the 184th elongated endodermal cell) root development stages. The Casparin strips were stained with basic fuchsin, and the results indicated that maize Casparin strip starts with a discontinuous lignin deposition, and then develops into ring like mature Casparian strips. The arrows indicated the dot-like staining of lignin at the endodermal CS domain. en, endodermis. Bars in **b** and **c** respectively equaled to 100 and 50 μm .

Figure 4 Lacking ZmESBL function leads to incomplete and less organized Casparian strips. **(a, b)** Lignin staining-based observation of Casparian strips in *ZmESBL^{crispr}* and wild-type plants grown under control and salinity conditions. Basic fuchsin staining of lignin (see methods) were performed, and the longitudinal view of the endodermal cells at 1.0 cm, 1.25 cm, 1.5 cm, 1.75 cm, 2.0 cm, 2.5 cm, and 3.0 cm from the root apex were performed **(a)**, and the fluorescence intensities were measure using the software Image J **(b)**. **(c)** Electron microscopy-based assay of Casparian strip structure in the wild-type and *ZmESBL^{crispr-1}* plants under control and salt conditions. **(d)** The widths of the later Casparian strip in the wild-type roots under control and salt conditions. **(e, f)** The effects of salinity stress on the transcript levels of *ZmESBL* in the total shoot or root tissue **(e)**, or in indicated segments of the primary root with six hours treatment. The results were means \pm s.d. of three independent experiments. Bars in **a** and **b** respectively equaled to 20 and 0.5 μ m.

(3) Do the shoots develop toxic symptoms even in high humidity if the salt treatment is prolonged? Can you really justify call this transpiration dependent?

Our response: We thank the reviewer for the comments. We had tried to grow the plant for three weeks, and we didn't observed apparent toxic symptoms. We believe it is reasonable to call this transpiration dependent.

(4) L249. Lack of ESBL increases Na^+ transport. Previous studies (Pfister et al 2014, eLife, Barberon et al 2016 Cell) suggest that defective CS leads to loss of K^+ and that this is the origin of the problems the plants have. The phenotype you should (chlorosis) is a classic K^+ deficiency phenotype. It would be ideal if you also included a measure of K in the tissues.

Our response: We thank the reviewer for this comment. To address this comments, we added the shoot and root K^+ content (wild type versus *ZmESBL^{crispr}* mutants) in the revised manuscript (**Supplementary Figure 10**). The results indicated that lacking ZmESBL led to significant reduction of K^+ content under both control and salt conditions, and under both high and low humidity conditions (**Supplementary Figure 10a and b**), with the salt-grown *ZmESBL^{crispr}* plants under low RH (50%) condition showed ~17% less shoot K^+ that grown under high RH (95%) conditions (**Supplementary Figure 10b**), which might made a contribution to the transpiration-dependent salt hypersensitivity of *ZmESBL^{crispr}* mutants. Also, we cited the mention studies (Pfister et al 2014, eLife, Barberon et al 2016 Cell).

Supplementary Figure 10 The root and shoot K^+ contents in the wild-type and *ZmESBL^{crispr}* plants. The plants were under the indicated conditions, and the results were means \pm s.d. of three independent experiments. Statistical significances were determined by a two-sided *t*-test.

(5) L260. It is curious that there is not more Na^+ in WT roots, as they exclude it from the transpiration stream. This is typically seen in roots of excluders. Why not here?
 L269. Even worse, now there is more Na^+ in mutant roots? Where does all the Na^+ go that WT excludes from shoot? It is shown that the more Na in mutant is in stele (Fig 5e), but why is there not more in WT in cortex if it is excluded by the endodermis? Where does it go?

Washed out during preparation? Why did it not get washed out of xylem vessels in the mutant then in the stele preparation (also apoplast)?

Our response: We thank the reviewer for this comment. We understand why the reviewer raised the concern that “*It is curious that there is not more Na⁺ in WT roots*”. Previous studies have shown the Na⁺ exclusion from the transpiration stream often leads to an increased Na⁺ concentration in root tissue, for instance, lacking of maize HKT and HAK family transporter increase the shoot Na⁺ contents but decrease root Na⁺ content (Zhang et al., Nat Plants. 2019, 5:1297-1308). In contrast, in this study, we observed that *ZmESBL^{crispr}* mutants showed an increase shoot and root Na⁺ content (**Figure 5f,h**). These results have been obtained in three independent experiments.

The discrepancy is likely because of, the Na⁺ excluded from the xylem vessels by the Na⁺ transporters were located in root stele, whereas the Na⁺ excluded by Casparin strip were located outside endodermis, where the Na⁺ is easier (as compared with the stele Na⁺) to be transported back to the environmental solutions either by symplastic or apoplastic mechanisms, and results in an insignificant increase of cortex Na⁺ in wild type as compared with that of *ZmESBL^{crispr}* mutants (**Figure 5e**; L260).

The excessive stele-Na⁺ loading of *ZmESBL^{crispr}* mutants increase the apoplastic loading of Na⁺ into root stele (as showed in Figure 5e), then increase the Na⁺ concentration in xylem vessel (as showed in **Figure 5e**), which in turn leads to excessive root-to-shoot Na⁺ delivery (as showed in **Figure 5h**), meanwhile, the increased Na⁺ concentration in xylem vessel also leading to an increased xylem-Na⁺ exclusion (by ZmHKT1 and ZmHAK4) into parenchyma cell in root stele, these Na⁺ retained in root stele and thereby leads to an increased root Na⁺ concentrations (as showed in **Figure 5h**).

During the sample preparation, the root samples were briefly rinsed with ddH₂O before the sample preparation, and then the root stele and outer part tissues were separated under condition without solution, hence it remains possible that the brief rinse has an effect on the apoplastic Na⁺ the sample preparation processes unlikely washed out the Na⁺ from the samples.

(6) L192. Speculation that ZmESBL is important for Casparian strip formation, as the first thought. But this makes no sense to me. Why not suberin? Why look at lignin as the first choice?

Our response: We thank the reviewer for this comment. We now added the assay of the suberin for the wild-type and *ZmESBL^{crispr}* mutants (**Supplementary Figure 7**), and described the suberin staining result in parallel with the Casparian strip assay in the revised manuscript. The results showed that both the wild-type and *ZmESBL^{crispr}* plants started endodermal suberin deposition at ~5.0 cm (about the 344th fully elongated endodermal cell), and the intensities of the staining florescence were comparable (**Supplementary Figure 7**), indicated that ZmESBL is unlikely associated with the regulation of endodermal suberin deposition.

Supplementary Figure 7 Comparison of suberin deposition in the wild-type and *ZmESBL^{crispr}* plant. **(a)** Fluorescence assay of endodermal suberin deposition using fluorol yellow 088 staining. **(b)** A graphical demonstration of location where the endodermal suberin deposition starts. en, endodermis. Bars equaled to 40 μm .

(7) L210. The use of the lignin stain as showing less lignin quantity. This should be stated much more carefully, as it is not trivial to equal stain signal with lignin quantity. Can lignin actually be properly quantified in the tissue rather than just using image analysis through biochemical or other means. Is the quantification of lignin in the tissues robust enough?

Our response: We thank the reviewer for this comment. To address this comment, we have measured the lignin contents in the primary root (up to 2.5 cm from the root apex) of the wild-type and *ZmESBL^{crispr}* plants grown under control conditions (**Supplementary Figure 8**). The results indicated that, the lignin contents in *ZmESBL^{crispr}* were $\sim 20\%$ lower than that of the wild type plants under both control and salt condition, with salt stress increased the lignin contents by 20% both in wild type and *ZmESBL^{crispr}* plants. The results have been added into the revised manuscript.

Supplementary Figure 8 Comparison of lignin content in the wild-type and *ZmESBL^{crispr}* plants under control and salt conditions. The plants with indicated genotypes were grown under control and salt (100 mM NaCl) conditions for 7 days, then the root tip segment (up to 2.5 cm from the root apex) of the primary root was collected to measure lignin content. The results were means \pm s.d. of three independent experiments.

(8) L200, use of *ESB1* promoter. Why not the *ESBL* promoter, the homologue of the *ZmESBL*? It makes little sense. In particular, as using the *ESB1* promoter ensures expression of *ZmESBL* in endodermis only.

Our response: We thank the reviewer for this comment. The original goal of this experiment is to determine whether *ZmESBL* proteins can be located to Casparian strip, hence we used the *pESB1* that is a well-known endodermis specific promoter. Following this suggestion, we have generated *pAtESBL-GFP-ZmESBL* plant, then analyzed the pattern of GFP-*ZmESBL* both in *pAtESB1-GFP-ZmESBL* and *pAtESBL-GFP-ZmESBL*, consequently observed that GFP-*ZmESBL* were enriched in Casparian strip domain in both transgenic plants (**Figure 3e; Supplementary Figure 5**).

Figure 3e. The subcellular location of GFP-*ZmESBL* protein in the Arabidopsis plants transformed with *pAtESB1-GFP-ZmESBL*. The red signal was basic fuchsin staining of lignin.

Supplementary Figure 5 The subcellular localization of GFP-ZmESBL1 in Arabidopsis endodermal cells. The subcellular location of GFP-ZmESBL protein in the Arabidopsis plants transformed with *pESB1-GFP-ZmESBL* (**a**) and *pESBL-GFP-ZmESBL* (**b**). The red signal indicated the basic fuchsin staining of lignin, and the yellow color indicated the localization of GFP-ZmESBL and lignin staining in endodermal Casparin strips. Bars = 30 μm .

(9) Paragraph starting L203. Clarification about cell length is needed here – it would be good to have a supp graph with the data for this paragraph.

Our response: We thank the reviewer for this valuable comment. We have analyzed the size of meristem, the cell length within each of the root segments that have been subjected to the analysis of Casparin strip, and we have added the result in the manuscript as a supplementary figure (**Supplementary Figure 6; see above response to the Major comment #2**).

(10) L351, daisy mutants, I don't understand why the authors say the finding support their data/hypothesis. The daisy mutants are salt sensitive, but contain similar Na^+ than wildtype, while they have defects in suberin deposition. How does this fit?

Our response: We thank the reviewer for this comment. It seems there was a misunderstanding of our results. As seen by the reviewer that the *daisy* mutants and wild-type plants contain similar Na^+ contents (**Figure 7f**), meanwhile, our data also showed that the *daisy* mutants showed a salt sensitivity comparable with wild-type plants (not hypersensitive) (**Figure 7d and e**), thereby supporting our conclusion that suberin deposition was less relevant to plant salt tolerance.

(11) Fig 3e. Image not acceptable. Image stack needs to be taken, and maximum intensity projection shown that displays the “net” of the CS (like shown in many of the references they give). Is some form of co-staining required here too? PI/Co-staining

Our response: We thank the reviewer for this comment. This comment has been fully addressed in our revised manuscript. Please see our response to your major Comments #8.

Minor

(1) Reference numbering is not according to how they appear in the text, but numbers plus alphabetic, is this normal?

Our response: We thank the reviewer for this comment. We have corrected the references in the revised manuscripts.

(2)L196, no reference given for “previous study”

Our response: The reference has been given in the revised manuscript (**Hosmani et al., 2013, Proc Natl Acad Sci, USA 110: 14498-14503**).

(3)L246, what does it mean ESBL “enhances” maize endodermal CS function. Enhances? Just above it was essential for it?

Our response: Thanks for the comments. In the revised manuscript, we have shown that the discontinuous and continuous staining of CS lignin was detectable at the younger endodermis cells under salt condition (**Figure 4a**). In addition, we observed by transmission electron microscopy that the later Casparian strips of the salt grown wild-type plants were 25% wider than that of the controls (**Figure 4d**). These results indicated that salt stress not only accelerates the development of Casparian strip, but also leads to a wider later Casparian strip. We have reworded the sentence into “These observations was in consistent with above observations that salt accelerates the development of Casparian strip and leads to a wider Casparian strip, and the salt-induced reprogramming of CS development is dependent upon the function of ZmESBL.

(4) L290. Extracting quantification data from staining again, need to be more carefully written.

Our response: Thanks for the comments. To address this comment, we have conducted further experiments to compare the Casparian strip of the wild-type, *esbl*, and *esbl*, including (1) quantification of CS autofluorescence, (2) analyzed the pattern of lignin deposition in CS domain. The results indicated that *esbl* mutant showed a decrease intensity of CS autofluorescence, and also a changed pattern of lignin deposition at endodermis (**Figure 6a and b**). These results have been added into the revised manuscript.

Figure 6a-b: (a) Confocal z-projections of endodermal Casparian strip autofluorescence. (b) The intensities and pattern of the CS autofluorescence in the indicated genotypes.

(5) L297, images might suggest that *esbl* mutants have more passage cells? Is this the case or just by chance like this in the given images?

Our response: Thanks for the comments. We have carefully checked our data, which suggested that the wild-type and *esbl* mutant conferred comparable number of passage cell. To avoid misleading, we have replaced the wild-type image in the revised manuscript.

(6) L317 *esbl* knockout has transpiration dependent phenotype. *Arabidopsis esbl* was shown to be salt sensitive under standard growth condition. Do the authors suggest that salt phenotype of *esbl* is also transpiration dependent?

Our response: Thanks for the comments. We indeed observed that salt-induced reduction of leaf SPAD value, and increase of shoot Na^+ content observed in *esbl* both dependent upon the transpiring environment (**Figure 7a-c**), hence suggest that salt hypersensitive phenotype of *esbl* is also transpiration dependent. Given the fact that *esbl* also confers a defective Casparian strip, these results were also consistent with our conclusion that CS barrier promotes shoot Na^+ exclusion and salt tolerance under transpiring condition.

(7) L398. Na^+ is the most abundant element in saline farmland. Is this a fact for each single saline farmland? Maybe re-write

Our response: Thanks for the comments. We have reworded the sentence into “Sodium (Na^+) is the most abundant soluble cation in many saline farmlands”.

(8) Fig 1, please clarify what a SPAD value is, it is not explained anywhere in the manuscript.

Our response: The revised manuscript has clarified the SPAD (soil plant analysis development; chlorophyll meter) in the legend of Figure 1 and in main text.

Reviewer #3 (Remarks to the Author):

To understand transpiration-dependent salt tolerance mechanism in crop, the authors screened maize inbred lines and identified CIMBL45, which shows transpiration-dependent hypersensitivity of salinity. Through the mapping, sequencing analysis and knock out studies, the authors discovered that a DIR family gene, ZmESBL, was the causative mutation and further analyzed its molecular mechanisms involved in Casparian strip development. The authors provided significant advances in the mechanisms of salt tolerance in crops and suggested a novel mechanism of ESBL in the regulation of Casparian strip development. However, some critical experiments/points should be considered in order to solidify the conclusion in the manuscript.

Our response: Thanks the reviewer for the comments, please see below our response to each of your comments.

1) Casparian strip structure in maize:

(1-1) It is necessary to clearly show the Casparian strip (CS) structure in maize, including how CS initiated in the early stages and matured in the later stage. CS of Arabidopsis exists in a ring-like structure, is it similar in maize? The CS of Arabidopsis starts with a dotted pattern and then develops in a continuous form through elongation. Is this similarity in maize?

Our response: We thank the reviewer for these valuable comments. In the revised manuscript, we have carefully analyzed the root meristem zone, elongation zone and mature zone of maize root (**Supplementary Figure 6**), which then permit us to describe the developmental process of endodermal Casparian strip accurately. showed that the discontinuous lignin deposition at the CS domain starts at 1.5 cm to the root apex (about the 64th elongated endodermal cell) in the control wild-type root, and then development into ring like continuous Casparin strips at 2.5 cm from the root apex (about the 144th elongated endodermal cell) (**Supplementary Figure 6; Figure 4a**). Such a developmental process is similar to that of Arabidopsis.

Supplementary Figure 6 The profiles of maize primary root under salt and control conditions. **(a)** A graphical demonstration of the profiles of the maize primary root grown under control condition. The red dash lines indicated Casparian strip, and the solid red lines indicated the fully functional CS barrier. **(b)** A demonstration of the cell size at indicated developmental stages. Calcofluor White stains cellulose. **(c-e)** The size of meristem **(c)** and elongation zone **(d)**, and the length of the fully elongated endodermal cell **(e)**. The genotypes and treatments were as indicated, and the result indicated that the salt (100 mM NaCl) reduced the size of the meristem by $\sim 500 \mu\text{m}$, reduced the length of the elongation zone by $\sim 500 \mu\text{m}$, and reduced the length of the elongated endodermal cells by 12%. **(f)** Longitudinal and cross section view of the Casparin strips at the early (2.0 cm to root apex; \sim the 104th elongated endodermal cell) and later (3.0 cm to root apex; \sim the 184th elongated endodermal cell) root development stages. The Casparin strips were stained with basic fuchsin, and the results indicated that maize Casparin strip starts with a discontinuous lignin deposition, and then develops into ring like mature Casparian strips. The arrows indicated the dot-like staining of lignin at the endodermal CS domain. en, endodermis. Bars in **b** and **c** respectively equaled to 100 and 50 μm .

(1-2) When and in what form does suberin deposition occur in maize?

Our response: We thank the reviewer for these valuable comments. In the revised manuscript, we showed that the suberin deposition starts at ~5 cm to root apex in the wild-type root, and *ZmESBL^{crispr}* and wild-type show comparable start position and pattern of suberin deposition (**Supplementary Figure 7**).

Supplementary Figure 7 Comparison of suberin deposition in the wild-type and *ZmESBL^{crispr}* plant. **(a)** Fluorescence assay of endodermal suberin deposition using fluorol yellow 088 staining. **(b)** A graphical demonstration of location where the endodermal suberin deposition starts. en, endodermis. Bars equaled to 40 μ m.

(1-3) How do the structural properties of CS change after salt treatment? Fig4 images are not enough to understand the structure of CS. It is also necessary to more accurately describe the ko phenotype in both quantitative and qualitative terms. Interpreting CS development in response of salt treatment have to be careful. Since CS development is related to the differentiation of the roots, it is necessary to carefully observe whether the root development is affected by the salt or whether the root development is the same, but only the CS

development is accelerated. CS position data should be analyzed after carefully observing changes in meristem zone size and elongation zone size.

Our response: We thank the reviewer for this important comment. To address this comment, we have carefully compared the root meristem zone size, elongation zone size and the length of the mature endodermis cell between wild-type and *ZmESBL^{crispr}* plants and under both control and salt conditions (**Supplementary Figure 6c-e**). The results indicated that, in the primary root of the wild-type plant, salt reduced the length of the meristem zone (from ~4.0 mm to ~3.5 mm) and the elongation zone (from ~3.0 mm to ~2.5 mm), and also reduced the length of the fully elongated endodermal cell from ~125 μm to 110 μm (**Supplementary Figure 6c-e**). We have also made a detail analysis of the effect of salt stress on CS onset, formation and structure, and observed that salt stress accelerate the development of Casparian strip (the analysis have taken the size of meristem zone, elongation zone, and the length of the cells in consideration). In addition, we compared the Casparian strip of the wild-type and *ZmESBL^{crispr}* mutant plants using transmission electron microscopy and additional confocal microscopy assay, the results showed that the mutants confer defective Casparian strips even in later stage endodermis cells (3.0 cm to the root apex; ~ the 184th elongated endodermal cell) under both control and salt conditions (**Figure 4c**), featured by less lignin deposition and discontinuous of the Casparian strips (**Figure 4a,b**). Moreover, we showed that the salt stress increased the width of the CS barrier by 20% in wild-type plants (**Figure 4d**). All these data have been added into the revised manuscript.

Supplementary Figure 6c-e The size of meristem (**c**) and elongation (**d**) zone, and the length of the fully elongated endodermal cell (**e**). The genotypes and treatments were as indicated, and the result indicated that the salt (100 mM NaCl) reduced the size of the meristem by ~500 μm , reduced the length of the elongation zone by ~500 μm , and reduced the length of the elongated endodermal cells by 12%. These inhibitions occurred both in wild-type and *ZmESBL^{crispr}* plants.

Figure 4 Lacking ZmESBL function leads to incomplete and less organized Casparian strips. **(a, b)** Lignin staining-based observation of Casparian strips in *ZmESBL^{crispr}* and wild-type plants grown under control and salinity conditions. Basic fuchsin staining of lignin (see methods) were performed, and the longitudinal view of the endodermal cells at 1.0 cm, 1.25 cm, 1.5 cm, 1.75 cm, 2.0 cm, 2.5 cm, and 3.0 cm from the root apex were performed **(a)**, and the fluorescence intensities were measured using the software Image J **(b)**. **(c)** Electron microscopy-based assay of Casparian strip structure in the wild-type and *ZmESBL^{crispr-1}* plants under control salt conditions. **(d)** The widths of the later Casparian strip in the wild-type roots under control and salt conditions.

(1-4) To verify CS permeability, PI was treated for 24 hours and then sectioned and observed. A control experiment for this PI method and quantitative validation is required. Lignin synthesis inhibitor, Lignin synthesis inhibitor+monolignol treatment can be used. To visualize permeability, longitudinal sections are required.

Our response: We thank the reviewer for these valuable comments. Follow the suggestion from the reviewer, we have also used lignin synthesis inhibitor (PA) to stop the formation of Casparian strip. Nevertheless, while it worked very well in Arabidopsis with a concentration of **10 μM** , the inhibitor cannot successfully stop the formation of Casparian strip with concentrations ranged from 10 to 200 μM , and treatments for 24 to 96 hours. This is probably due to that the multilayered (~6) cortex cells prevent the inhibitor to reach endodermis.

We have visualized permeability with the longitudinal sections, and the results indicated that the longitudinal and transvers sections led to the same conclusion (**Figure 5a,b**). In addition, as a verification of the accuracy of our PI penetration assay, we showed that the PI penetration only been blocking at the root zone where the later Casparian strip has been formed (2.5 cm from the root tip; ~144th elongated endodermal cell) (**see Figure 4c**). These analyses can justify the PI penetration assay used in this study.

Figure 5a PI penetration-based assay of the function of endodermal CS barrier in primary roots. PI penetration assay was performed at 1.5 cm, 2.0 cm, 2.5 cm, 3.0 cm, and 3.5 cm from the root apex (see methods).

2) Molecular mechanisms of *ZmESBL*:

(2-1) The authors suggest that the action of *ZmESBL* is different from that of *ESBL1*, but there is no suggestion as to what mechanism they think it is. Is it linked to a decrease in lignin production? Does the ectopic treatment of monolignol restore the *ko* phenotype?

Our response: We thank the reviewer for the comment. Follow the suggestion from the reviewer, we have measured the lignin content in the wild-type and *ZmESBL^{crispr}* plants, the results indicated that *ZmESBL^{crispr}* showed a ~20% reduction of lignin content under both control and salt conditions (**Supplementary Figure 8**). Nevertheless, we observed that the monolignol (20 μM coniferyl alcohol plus 20 μM sinapyl alcohol) treatment cannot rescue the Casparian strip structure and function in Arabidopsis *esbl* mutant (**Supplementary Figure 12**). These results indicated that the defective Casparian strip causing by lacking of ESBL orthologs is unlikely ascribed to the deficiency of monolignol biosynthesis. These data have been added into the revised manuscript.

Supplementary Figure 8 Comparison of lignin content in the wild-type and *ZmESBL^{crispr}* plants under control and salt conditions. The plant with indicated genotypes were grown under control and salt (100 mM NaCl) conditions for 7 days, then the root tip segment (up to 2.0 cm from the root apex) of the primary root were collected to measure lignin content. The results in **b** were means \pm s.d. of three independent experiments.

Supplementary Figure 12 The effect of monolignol treatment on the endodermal CS barriers in the wild-type and *esbl-1* plants. **(a)** Confocal z-projections of endodermal Casparian strip autofluorescence in the primary root of wild-type and *esbl-1* plants with the indicated treatments. **(b)** PI penetration-based assay of the function of the endodermal CS barrier in indicated samples. The images were taken at $\sim 25^{\text{th}}$ endodermal cell after the onset of elongation. The results indicated that the monolignol (20 μm coniferyl alcohol plus 20 μm sinapyl alcohol) application can neither restore the CS structure **(a)** nor rescue the function of the endodermal barrier **(b)** in *esbl-1* mutant. Bars = 20 μm .

(2-2) Or is it related to the *CIF1/2-SGN3* pathway? Does ectopic treatment of *CIF1/2* restore the *ko* phenotype? How about the localization of *CASP1* in *ko*? Further experimentation is needed to see how *ZmESBL* affects the already identified critical elements of CS development.

Our response: We thank the reviewer for the comment. Follow the suggestion from the reviewer, we generated a pCASP1-CASP1-GFP in *esbl* mutant background, and observed that neither the formation of the discontinuous CASP1 scaffolds at the onset stage nor the formation of the continuous CASP1 scaffolds at the later stage of CS formation was affected in *esbl* mutant (**Supplementary Figure 13**). These observations suggested that ESBL unlikely involves in the early (onset stage) of CS formation. We suggested that ESBL might be part of the machinery that is recruited to the CS domain to mediate a later process of CS formation. Given the fact that the DIR family proteins have been shown to mediate regio- and stereoselectivity of bimolecular coupling during lignan and lignin biosynthesis, it remains highly possible that ESBL ortholog mediate bimolecular coupling during the polymerization of monolignol.

Supplementary Figure 13 The subcellular localization of CASP1-GFP in the wild-type and *esbl-1* background. The discontinuous (a) and continuous (b) CASP1-GFP were observed in both wild-type and *esbl-1* plants, and were observed at the same root developmental stage reflecting by the number of endodermal cells from the onset of cell elongation (c,d). Bars = 20 μ m.

We have also analyzed the effect of CIF treatment on the CS structure and barrier function, and observed that the CIF treatment can restore the function of the endodermal

barrier, nevertheless, the functional rescue is likely associated with the ectopic lignification at the of extra CS at the endodermal cell-cell contact sites (**see the Figure bellow**), rather than rescue the formation of a normal CS barrier. As these results didn't provide additional information for the understanding of the ESBL function, the results haven't been added into the revised manuscript.

Figure for comment #2-2 The effect of CIF treatment on the CS structure and barrier function. The results indicated that CIF treatment led to an ectopic deposition of lignin at the cortex face of the endodermal cell-cell contact site (indicated by red arrow), but cannot restore the lignin deposition at the normal CS formation site (indicated by yellow arrow). The result also showed that CIF treatment can rescue the barrier role of endodermis, probably due to the ectopic deposition of lignin. Bars = 20 μm .

(2-3) *The authors tried to understand the mechanism of ESBL using the Arabidopsis mutant study, which can be improved by more accurately describing the phenotype and adding quantitative assessments. “Weak” or “strong” is not enough to describe CS structure. It is necessary to describe the characteristics by observing the CS structure in a more magnified view (Fig 6a), and it is necessary to describe the suberin pattern in comparison with the development stage (Fig 6b). Both are missing quantitative analysis.*

Our response: We appreciate the reviewer for this valuable suggestion. In the revised manuscript, we have analyzed the CS structure in a more magnified view, and have quantified the intensity and investigated the pattern of the lignin deposition in endodermis (**Figure 6a,b**). In addition, we have compared the suberin pattern in the scenario of development stage

(Figure 6c), and have conducted a quantitative analysis of the suberin deposition (Figure 6e), the results indicated that maize *ZmESBL^{crispr}* and Arabidopsis *esbl* mutant display a suberin deposition pattern comparable with their wild-type plant. These results were in consistent with our conclusion that *ZmESBL* and Arabidopsis *ESBL* are unlikely associated with the deposition of suberin in endodermal cells.

Figure 6 Arabidopsis *ESBL* (the ortholog of *ZmESBL*) is essential for CS development and salt tolerance under high transpiring condition. (a) Confocal z-projections of endodermal Casparian strip autofluorescence. (b) The intensities and pattern of the CS autofluorescence in the indicated genotypes. The measurements were conducted at the locations indicated by the yellow lines in a. (c-e) Fluorescence assay of endodermal suberin deposition using fluorol yellow 088 staining, showed the locations where the endodermal suberin deposition starts (c), the pattern (d) and intensities (e) of fluorescence detected in the indicated samples.

3) Minor points:

(1) Description of SPAD is missing.

Our response: We have clarified SPAD (soil plant analysis development; chlorophyll meter) in the legend of Figure 1 and in main text.

(2) Localization of *ZmESBL* (Fig 3e): It is good to clearly show correlation of Casparian

strip. Additionally, using PM reporter protein with ZmESBL helps readers understand its localization pattern.

Our response: Thanks the reviewer for the comments. We have provided photos showed the co-localization of ZmESBL-GFP and basic fuchsin staining of lignin in the Casparian strip domain the revised manuscript (**Figure 3e; Supplementary Figure 5**).

Figure 3e. The subcellular location of GFP-ZmESBL protein in the Arabidopsis plants transformed with *pAtESB1-GFP-ZmESBL*. The red signal was basic fuchsin staining of lignin.

Supplementary Figure 5 The subcellular localization of GFP-ZmESBL1 in Arabidopsis endodermal cells. The subcellular location of GFP-ZmESBL protein in the Arabidopsis plants transformed with *pESB1-GFP-ZmESBL* (a) and *pESBL-GFP-ZmESBL* (b). The red signal indicated the basic fuchsin staining of lignin, and the yellow color indicated the localization of GFP-ZmESBL and lignin staining in endodermal Casparian strips. Bars = 30 μ m.

(3) In 75, 214 lines, the author said that CS shows large plasticity, but since the degree and range of plasticity of CS and suberin are different, it is necessary to distinguish and explain it.

Our response: Thanks the reviewer for the comments. In the revised manuscript, we distinguish and explain the degree and range of plasticity of CS and suberin by adding “*For instance, the biosynthesis and degradation of endodermal suberin lamellae is modulated by various plant hormones (e.g. ABA and ethylene) in response to nutrient stresses (Barberon et al., 2016, Cell 164: 447-59), and the endodermal CS barrier continuity is constantly checked by two CIF peptides and its maturation can be accelerated by low-K⁺ signaling (Wang et al., Dev Cell. 2021, 56:781-794; Alassimone et al., 2016; Doblus et al., 2017; see L75).*”.

(4) In this paper, the roles of CS lignin and suberin are distinguished, and the importance of CS is especially emphasized. The characteristics of the two and the molecular mechanism of their action need to be explained.

Our response: Thanks the reviewer for the comments. In the revised manuscript, we explained the molecular mechanisms of CS lignin and suberin by saying “*The maturation of endodermis cells is composed of the lignin deposition at Casparin strip domain at the early stage and the formation of suberin lamella at the later stage^{19,40}, with lignin-based CS barrier prevents apoplastic delivery of minerals², and suberin lamella establishes a barrier for uptake of minerals from the apoplast in to the endodermis³.*” in the discussion section.

Reviewers' Comments:

Reviewer #1:

Remarks to the Author:

The authors have addressed the minor comments I originally communicated. It is an excellent contribution to the field.

Reviewer #2:

Remarks to the Author:

The authors have satisfactorily addressed my previous comments in the main.

My criticism of any significance would be that measurement of fluorescence for lignin quantification is likely not to be the most reliable technique. It is qualitative at best. It would have been better to make a biochemical quantification.

L202- Use of the Arabidopsis ESB and ESBL promoters to drive expression at the CS surely does not as this stage test localisation – more function, and then this can later be used as evidence for localisation. I would recommend rewording.

L394 The phenotype of Arabidopsis esbl plants is also likely due to K deficiency - not measured.

Multiple minor grammar issues throughout.

Reviewer #3:

Remarks to the Author:

The manuscript has been significantly improved and has fully addressed all concerns raised.

The following outlines our specific responses to the reviewers' comments:

Reviewer #1 (Remarks to the Author):

The authors have addressed the minor comments I originally communicated. It is an excellent contribution to the field.

Our response: We thank the reviewer for the positive feedback on our submission.

Reviewer #2 (Remarks to the Author):

The authors have satisfactorily addressed my previous comments in the main.

1. My criticism of any significance would be that measurement of fluorescence for lignin quantification is likely not to be the most reliable technique. It is qualitative at best. It would have been better to make a biochemical quantification.

Our response: We thank the reviewer for this comment. To address this comment, we have measured the lignin content of maize *ZmESBL*^{crispr} mutants, Arabidopsis *esbl* mutants and the wild-type controls, and results indicated that the mutants showed significant reduction of lignin content as compared with the wild-type controls. The results have been included in the revised manuscript (Supplementary Figure 8 and Supplementary Figure 12).

Supplementary Figure 8 The lignin content of the wild-type and *ZmESBL*^{crispr} plants under the control and salt conditions. The plants with the indicated genotypes were grown under control and salt (100 mM NaCl) conditions for 3 days, then the indicated root tip segment of the primary root were collected to measure the lignin content. The results were means \pm s.d. of three independent experiments. Statistical significances were determined by a two-sided *t*-test.

Supplementary Figure 12 The lignin content of the *esbl* and wild-type plants. The plants with the indicated genotypes were grown in 1/2 MS medium for 10 days, and then the root tissues were collected to measure the lignin content using the previously described method¹. The results were means \pm s.d. of 9 independent experiments. Statistical significances were determined by a two-sided *t*-test.

2. 202- Use of the Arabidopsis ESB and ESBL promoters to drive expression at the CS surely does not as this stage test localisation – more function, and then this can later be used as evidence for localisation. I would recommend rewording.

Our response: We thank the reviewer for this comment. The logic for this paragraph is to analyze the expression pattern and subcellular localization of ZmESBL to suggest that ZmESBL is likely to play a role in the regulation of CS formation. To address the comment, we have reworded this paragraph into “To address this hypothesis, firstly, we analyzed the tissue specificity of *ZmESBL* expression, and observed that *ZmESBL* mainly expressed in root tissue (Figure 3b, c), and its transcripts were predominantly detected in the endodermis and the stele cells adjacent to the endodermis (Figure 3d). Secondly, we determined the subcellular localization of ZmESBL by generating and analyzing the Arabidopsis plants expressing GFP-ZmESBL under the control of the *ESBL* (the ortholog of *ZmESBL* in Arabidopsis) promoter and the endodermal-specific *ESB1* promoter³² (see methods). The results indicated that the GFP-ZmESBL was predominantly detected at the endodermal CS domain in both *pESBL-GFP-ZmESBL* and *pESB-GFP-ZmESBL* (Figure 3e; Supplementary Figure 5). The expression pattern and subcellular localization of ZmESBL1 suggested that ZmESBL likely plays a role in the regulation of endodermal Casparian strip formation and/or suberin deposition.)”

3. L394 The phenotype of Arabidopsis *esbl* plants is also likely due to K deficiency - not measured.

Our response: We thank the reviewer for this comment. We have measured the shoot K⁺ content of the *esbl* and wild-type plants, and the data have been included in the revised manuscript (Supplementary Figure 15).

Supplementary Figure 15 The shoot K⁺ contents of *esbl* and wild-type plants with the indicated treatments. The results were means \pm s.d. of three independent experiments. Statistical significances were determined by a two-sided *t*-test.

4. Multiple minor grammar issues throughout.

Our response: We thank the reviewer for this comment. We have carefully looked through the manuscript to correct the grammar issues.

Reviewer #3 (Remarks to the Author):

The manuscript has been significantly improved and has fully addressed all concerns raised.

Our response: We thank the reviewer for the positive feedback on our submission.

Reviewers' Comments:

Reviewer #2:

Remarks to the Author:

The authors have addressed most of my comments.

The manuscript still requires extensive copy editing for grammar.

I also advise caution when using the term Na⁺ homeostasis, it is overused and not always in the correct context. Have the authors, or indeed anyone shown that Na⁺ homeostasis exists. It is more often maintenance of cellular function in the context of raised Na⁺ concentrations.

● **Our response to the comments from the reviewer #2:**

(1) *The manuscript still requires extensive copy editing for grammar.*

Our response: We have carefully gone through the manuscripts to edit the grammar.

(2) *I also advise caution when using the term Na^+ homeostasis, it is overused and not always in the correct context. Have the authors, or indeed anyone shown that Na^+ homeostasis exists. It is more often maintenance of cellular function in the context of raised Na^+ concentrations.*

Our response: We thank the reviewer for the comment. We have reworded “ Na^+ homeostasis” into “ Na^+ concentrations”, “ Na^+ delivery” and “shoot Na^+ exclusion” (depend on the context) in the revised manuscript.

● **Our response to the comments from the editorial team:**

(1) The raw data for every image of this manuscript.

Our response: We have uploaded all the raw data (with replicates) for every image to goggle drive (<https://drive.google.com/drive/folders/1th1Babne0nngU08oPile29OTphRXisWm?usp=sharing>).

Meanwhile, we have submitted raw data and raw images of the Figures as a zipped file.

(2) There are a few mistakes in Figures, which are either noticed by the editorial team or observed during the revision. We have corrected these Figures in the revised manuscript.

(1) Correction of Fig. 4a: As I have mentioned in my previous emails, we revealed that a few of the images in the original Fig. 4a has been changed in the brightness inappropriately, therefore, we have arranged independent assays recently, and the overall result is in consistent with our previous conclusion. We have replaced Fig. 4a and Fig. 4b (the fluorescence intensities of Fig. 4a) with the new data in the revised Manuscript.

The revised Fig. 4a-b

(2) Correction of Fig. S7a: We observed that the image (4-5 cm to root apex/*ZmESBLcrispr-2*) is an incorrect image. We have replaced the image with the correct one in the revised manuscript.

The revised Supplementary Fig. 7a

(3) Correction of Fig. 7: We found that the “labels” for the treatments (Control and Salt) were missing in Fig. 7a and Fig. 7d. We have added the “labels” in the revised manuscript.

With many thanks for considering our manuscript, I look forward to hearing from you in due course.

With best wishes,

Professor Caifu Jiang,
Department of Life Sciences,
China Agricultural University, China